# Cerebellar Purkinje cells control posture in larval zebrafish (*Danio rerio*)

Franziska Auer[1], Katherine Nardone[1], Koji Matsuda[2], Masahiko Hibi[2], David Schoppik[1]*

[1]Depts. of Otolaryngology, Neuroscience & Physiology, and the Neuroscience Institute, NYU Grossman School of Medicine, New York, United States; [2]Division of Biological Science, Graduate School of Science, Nagoya University, Nagoya, Japan

*For correspondence:
schoppik@gmail.com

Competing interest: The authors declare that no competing interests exist.

## eLife Assessment

This **important** study successfully applies an innovative chemogenetic tool to investigate cerebellar function to advance our understanding of the contributions of Purkinje cell populations to postural control in larval zebrafish. The evidence supporting the conclusions is **convincing** and supported by rigorous statistical analysis. The study highlights the power of combining genetically targeted perturbations with quantitative high-throughput behavioral analysis and original microscopy tools.

**Abstract** Cerebellar dysfunction leads to postural instability. Recent work in freely moving rodents has transformed investigations of cerebellar contributions to posture. However, the combined complexity of terrestrial locomotion and the rodent cerebellum motivate new approaches to perturb cerebellar function in simpler vertebrates. Here, we adapted a validated chemogenetic tool (TRPV1/capsaicin) to describe the role of Purkinje cells — the output neurons of the cerebellar cortex — as larval zebrafish swam freely in depth. We achieved both bidirectional control (activation and ablation) of Purkinje cells while performing quantitative high-throughput assessment of posture and locomotion. Activation modified postural control in the pitch (nose-up/nose-down) axis. Similarly, ablations disrupted pitch-axis posture and fin-body coordination responsible for climbs. Postural disruption was more widespread in older larvae, offering a window into emergent roles for the developing cerebellum in the control of posture. Finally, we found that activity in Purkinje cells could individually and collectively encode tilt direction, a key feature of postural control neurons. Our findings delineate an expected role for the cerebellum in postural control and vestibular sensation in larval zebrafish, establishing the validity of TRPV1/capsaicin-mediated perturbations in a simple, genetically tractable vertebrate. Moreover, by comparing the contributions of Purkinje cell ablations to posture in time, we uncover signatures of emerging cerebellar control of posture across early development. This work takes a major step towards understanding an ancestral role of the cerebellum in regulating postural maturation.

## Introduction

In vertebrates, cerebellar activity underlies proper posture, defined as the relative orientation of body parts in space (*Sprague and Chambers, 1953*, *Kleine et al., 2004*, *Ioffe, 2013*, *Tsutsumi et al., 2020*, *Becker and Person, 2019*, *Heiney et al., 2014*, *Bo et al., 2008*, *Darmohray et al., 2019*). The cerebellum integrates sensory information from vestibular (balance), visual, and proprioceptive systems (*Ioffe, 2013*). These sensations are transformed into precise and coordinated adjustments in muscle tone and contraction allowing animals to control posture (*Horak and Diener, 1994*). Disruptions to mature cerebellar function lead to instability, unsteady gait and a compromised sense of

balance (*Morton et al., 2010*). Development of the cerebellum coincides with postural maturation, and early development of the cerebellum has been extensively studied (*Sepp et al., 2024*, *Leto et al., 2016*, *Beckinghausen and Sillitoe, 2019*). Notably, changes to morphology and activity of the output neurons of the cerebellar cortex, Purkinje cells, are thought to underlie the gradual refinement of motor control (*Beekhof et al., 2021*). To date, the contributions of developing Purkinje cells to postural control remains poorly understood.

Kinematic quantification by pose estimation in rodents (*Machado et al., 2015*, *Sheppard et al., 2022*) has opened a window into cerebellar contributions to postural behaviors in health and disease (*Darmohray et al., 2019*, *Machado et al., 2020*, *Jaarsma et al., 2023*). However, terrestrial gait and locomotion are complex and especially difficult to study during development; tracking and analysis is often limited to measures such as the time of head elevation or the duration of walking stance (*Swann and Brumley, 2019*). In contrast, the biophysical challenges of maintaining posture underwater are straightforward to define (*Erich Von Holst, 1973*, *Sfakiotakis et al., 1999*, *Bagnall and Schoppik, 2018*). For example, larval zebrafish balance in the pitch axis (nose-up/nose-down) by timing locomotion to countermand gravity-induced destabilization (*Ehrlich and Schoppik, 2017a*, *Ehrlich and Schoppik, 2017b*) and by coordinated use of paired appendages (fins) and axial musculature (trunk; *Ehrlich and Schoppik, 2019*). The small size and rapid development of the larval zebrafish allow high-throughput measurements of postural control (i.e. pitch-axis kinematics and fin/trunk coordination) from freely swimming subjects (*Zhu et al., 2023*).

The larval zebrafish is a powerful model to investigate cerebellar development and function (*Pose-Méndez et al., 2023*). Anatomically, the zebrafish cerebellum shares the same circuit structure as the mammalian cerebellum (*Hibi and Shimizu, 2012*). The zebrafish cerebellum is compartmentalized into regions with distinct response properties and output targets (*Hibi and Shimizu, 2012*, *Heap et al., 2013*, *Matsui et al., 2014*, *Harmon et al., 2017*, *Dorigo et al., 2023*). Multimodal representations were found in both cerebellar granule cells (*Knogler et al., 2017*, *Sylvester et al., 2017*) and Purkinje cells (*Harmon et al., 2017*, *Chang et al., 2021*). Functional assays established a role for the larval zebrafish cerebellum in motor control, sensorimotor integration and predictive neural processing, particularly in response to visual input (*Ahrens et al., 2012*, *Sengupta and Thirumalai, 2015*, *Scalise et al., 2016*, *Knogler et al., 2019*, *Harmon et al., 2020*, *Lin et al., 2020*, *Prat et al., 2022*, *Najac et al., 2023*, *Narayanan et al., 2024*). Finally, brain-wide imaging studies have established balance-relevant sensitivity in the cerebellum, identifying neurons that encode body angle and velocity (*Migault et al., 2018*) and neurons responsive to direct inner-ear stimulation (*Favre-Bulle et al., 2018*). Overwhelmingly, this work has been done in reduced or restrained preparations, limiting insight into the cerebellar contribution to natural behaviors.

Powerful new opto- and chemogenetic (*Armbruster et al., 2007*) approaches allow control of particular cerebellar cell types, reviewed in *Prestori et al., 2020*. Recent work used such activation/inhibition to investigate cerebellar contributions to sensorimotor (*Heiney et al., 2021*, *Gaffield et al., 2022*, *Verpeut et al., 2023*, *Soetedjo and Horwitz, 2023*) and non-sensorimotor behaviors (*Carta et al., 2019*, *Chen et al., 2022*, *Jackman et al., 2020*, *Zamudio et al., 2023*) in health and disease (*Chao et al., 2020*, *Chao et al., 2021*, *van der Heijden et al., 2023*). Both approaches come with technical hurdles: optogenetics requires targeting light to the cerebellum, a particular challenge when untethered animals can move freely in depth, while chemogenetics uses bioactive co-factors (*Gomez et al., 2017*). A chemogenetic approach to cerebellar control with a non-bioactive ligand would be a welcome advance, particularly to study posture without visual interference (i.e. in the dark). One validated path forward is to express the rat non-selective cation channel TRPV1 and its ligand capsaicin in zebrafish (*Chen et al., 2016*). The endogenous zebrafish TRPV1 channel is capsaicin-insensitive (*Gau et al., 2013*), so targeted expression of rat TRPV1 allows cell-type-specific control: low-doses of capsaicin can activate sensory and hypothalamic neurons while high-doses are excitotoxic (*Chen et al., 2016*). Capsaicin can be dissolved in water and is readily absorbed by freely swimming larval zebrafish, sidestepping invasive procedures and the need for visible light. Finally, the conductance of a TRP channel is ~1000 x that of a channel rhodopsin (*Bernstein et al., 2012*) suggesting that even low levels of TRPV1 expression will be biologically effective.

Here, we used the TRPV1/capsaicin system to investigate the contribution of cerebellar Purkinje cells to postural behaviors as larval zebrafish swam freely in depth. Both activation and ablation of Purkinje cells could induce changes in pitch axis posture. Ablation in older larvae resulted in bigger

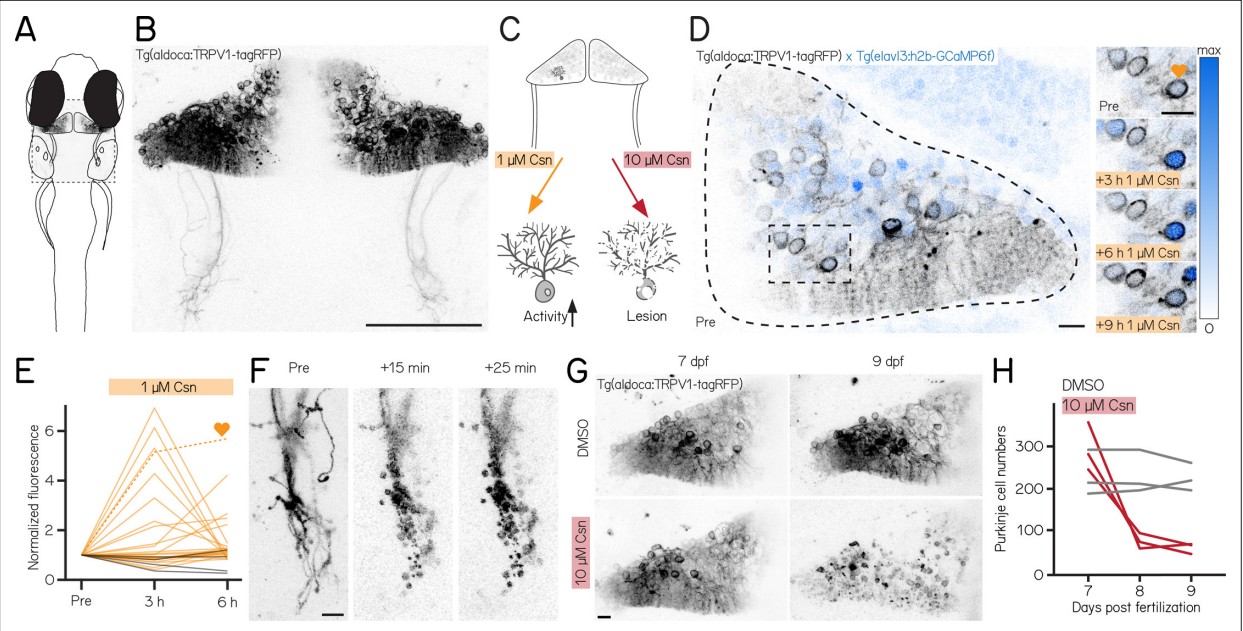

**Figure 1.** A chemogenetic approach allows dose-dependent activation and lesion of Purkinje cells in the cerebellum. (**A**) Schematic of a larval zebrafish overlaid with a confocal image of labeled Purkinje cells in the cerebellum. Gray rectangle corresponds to field of view in (**B**). (**B**) Confocal image of Purkinje cells in the cerebellum of a 7 days post-fertilization (dpf) *Tg(aldoca:TRPV1-tagRFP)* larvae. Scale bar 100 μm. (**C**) Schematic of strategy for dose-dependent activation (yellow, left) or lesion (red, right) of Purkinje cells by addition of the TRP channel agonist capsaicin (Csn). (**D**) Confocal image (inverted look-up table) of one cerebellar hemisphere of *Tg(aldoca:TRPV1-tagRFP)*; *Tg(elavl3:h2b-GCaMP6f)* larvae before, 3, 6, and 9 hr after addition of capsaicin. Heart corresponds to the labelled trace in (**E**). (**E**) Normalized change in fluorescence following treatment with 1 μM capsaicin in individual Purkinje cells as a function of time. Purkinje cells from *Tg(aldoca:TRPV1-tagRFP);Tg(elavl3:h2b-GCaMP6f)* larvae (orange) and *Tg(elavl3:h2b-GCaMP6f)* control larvae (grey). (**F**) Timelapse images of Purkinje cell axons in *Tg(aldoca:TRPV1-tagRFP)* larvae immediately after addition of 10 μM capsaicin. Scale bar 10 μm. (**G**) Confocal images of cerebellar hemispheres of *Tg(aldoca:TRPV1-tagRFP)* larvae before (7 dpf, left) and after (9 dpf, right) treatment with 10 μM capsaicin. Control larvae (DMSO, top) and lesion larvae (10 μM capsaicin, bottom). Scale bar 10 μm. (**H**) Quantification of Purkinje cell numbers of fish (n=3) from (**G**).

The online version of this article includes the following figure supplement(s) for figure 1:

**Figure supplement 1.** Chemogenetic activation of Purkinje cells is reversible.

disruptions to posture, allowing inference of the functional consequences of cerebellar development. Furthermore, ablation of Purkinje cells in older larvae disrupted the coordination of trunk and paired appendages (fins), impairing vertical navigation. Finally, we could reliably decode pitch-tilt direction from patterns of Purkinje cell activity. Taken together our results establish a clear role for the cerebellum in larval zebrafish postural control, even during the earliest stages of development. More broadly, our work establishes a new method to manipulate cerebellar output while performing quantitative high-throughput measures of unconstrained posture and locomotion. Our data are therefore a step towards defining an ancestral role for the highly conserved cerebellum in postural control.

## Results

### A new tool for chemogenetic activation or ablation of Purkinje cells

We used a new tool to control Purkinje cells: the transgenic line *Tg(aldoca:TRPV1-tagRFP)*. Fish in this line express rat TRPV1, a capsaicin-sensitive non-selective cation channel, exclusively in all cerebellar Purkinje cells (*Figure 1A and B*; *Tanabe et al., 2010*).

Endogenous zebrafish TRPV1 channels are insensitive to capsaicin (*Gau et al., 2013*). Previous descriptions of rat TRPV1 in zebrafish sensory and hypothalamic neurons establish dose-dependent chemogenetic manipulation (*Chen et al., 2016*). We expect low-doses of capsaicin to depolarize Purkinje cells (*Figure 1C*, left), while high-doses should be excitotoxic (*Figure 1D*, right).

First, we assayed capsaicin concentrations and incubation times to identify a dose that would achieve long-term depolarization without cell death. We co-expressed a nuclear-targeted calcium

indicator, GCaMP6f (*Figure 1D*) in all neurons using the *Tg(elavl3:h2B-GCaMP6f)* line for longitudinal imaging of neuronal activity. Previous work used 1 µM of capsaicin for long-term activation (*Chen et al., 2016*). We therefore imaged the cerebellum of *Tg(aldoca:TRPV1-tagRFP);Tg(elavl3:h2B-GCaMP6f)* fish prior to and 3, 6, and 9 hr after 1 µM capsaicin treatment (*Figure 1D*). We screened fish for comparable brightness and selected fish that had clearly visible expression but were not overly bright (Materials and methods).

Prolonged exposure to a low dose of capsaicin increased cerebellar activity (*Figure 1E*, *Figure 1—figure supplement 1A*). At each timepoint, TRPV1-expressing cells showed increased intensity relative to a pre-capsaicin baseline, while TRPV1-negative cells did not (*Figure 1E*, 3/6 hr post 1 µM capsaicin: 28%/20%/ TRPV1+ cells $F/F_0 > 2$; 40 cells from 3 fish vs. 0%/0%/ TRPV1- cells $F/F_0 > 2$; 44 cells from 4 fish; activated cells after 6 hr of capsaicin treatment: 0/44 TRPV1- vs. 8/40 TRPV1+; Fisher's exact test: p=0.0018).

Different cells showed increased activity at the 3,6, and 9 hr timepoints, and the same cells were differentially active at different timepoints. We interpret this as evidence that 1 µM of capsaicin could sporadically activate subsets of Purkinje cells. Notably, in one fish that had particularly strong tagRFP expression we observed a small number of neurons at the 9 hr timepoint with bright, speckled fluorescence suggestive of cell death (*Figure 1—figure supplement 1C*). We did not observe any signs of cell death at the 6 hr timepoints (6 TRPV1+ fish at 6 hr post 1 µM capsaicin). We therefore set an upper limit of 6 hr of exposure to 1 µM capsaicin for activation experiments.

Induced activation was reversible, even after prolonged exposure to 1 µM of capsaicin. We tested whether the elevated patterns of neuronal activity that we observed in the presence of capsaicin would return to baseline by imaging cerebellar Purkinje cells in *Tg(elavl3:h2B-GCaMP6f)* before exposure, after 6 hr of 1 µM capsaicin, and 40 min after washout. Relative to baseline, fluorescent intensities increased after 6 hr, as in *Figure 1E*. Importantly, fluorescence returned to baseline levels after 40 min of washout (*Figure 1—figure supplement 1B*, 6h post 1 µM capsaicin: 40.9%/ TRPV1+ cells $F/F_0 > 2$; washout: 0%/ TRPV1+ cells $F/F_0 > 2$; 22 cells from 3 fish; activated cells before 1 µM capsaicin treatment vs. after washout: Fisher's exact test: p=1). We conclude that capsaicin-induced activation is reversible after washout.

Exposure to high doses of capsaicin caused rapid axonal degeneration and cell death. We developed a protocol for Purkinje cell lesion: *Tg(aldoca:TRPV1-tagRFP)* larvae (without GCaMP6f) were imaged at 7 dpf, at 8 dpf after 1 hr of 10 µM capsaicin treatment and again at 9 dpf (*Figure 1G*). Timelapse imaging of the Purkinje cell axons showed rapid degeneration already 15 min after capsaicin treatment started (*Figure 1F*). Cell numbers rapidly declined after 1 hr of 10 µM capsaicin treatment and did not show any signs of recovery at 9 dpf (*Figure 1G and H*; median [inter-quartile range]; 7 dpf: control 213 [195 – 271] cells vs. pre lesion 282 [255 – 366] cells; 9 dpf: control 218 [201 – 250] cells vs. post lesion 68 [52 – 70] cells; 3 control and 3 lesioned fish).

Consistent with prior work in other cell populations (*Chen et al., 2016*), we found that chemogenetic use of the capsaicin/TRPV1 system can be used to reversibly activate or rapidly ablate cerebellar Purkinje cells in larval zebrafish.

## Purkinje cells regulate postural control in the pitch axis

We used our Scalable Apparatus to Measure Posture and Locomotion (SAMPL) to measure posture and locomotion in freely swimming zebrafish (*Zhu et al., 2023*). SAMPL is a high-throughput videographic approach that measures kinematic parameters of posture and locomotion from fish swimming in a predominantly vertical arena that encourages navigation in depth (*Figure 2A and B*) allowing us to analyze postural changes in the pitch (nose-up/nose-down) axis. Larval zebrafish locomote in discrete bouts of rapid translation (*Figure 2B*, grey lines). To navigate up/down, fish sequence these bouts while maintaining a nose-up/nose-down pitch (*Zhu et al., 2024*). Notably, climb/dive bouts are defined relative to the *trajectory* of the bout. Climb/dive bouts can therefore be initiated from either nose-up (positive) or nose-down (negative) *postures*.

Nose-up 'climb' bouts (*Figure 2E*) engage both axial musculature of the body and the fins to produce a net upward trajectory while nose-down 'dive' bouts (*Figure 2J*) rely on axial musculature alone and have a net downward trajectory (*Ehrlich and Schoppik, 2019*). Notably, postural angles after either climb or dive bouts tend to increase, a consequence of restorative rotations that counteract destabilizing torques (*Ehrlich and Schoppik, 2017b*). SAMPL's automated and high-throughput

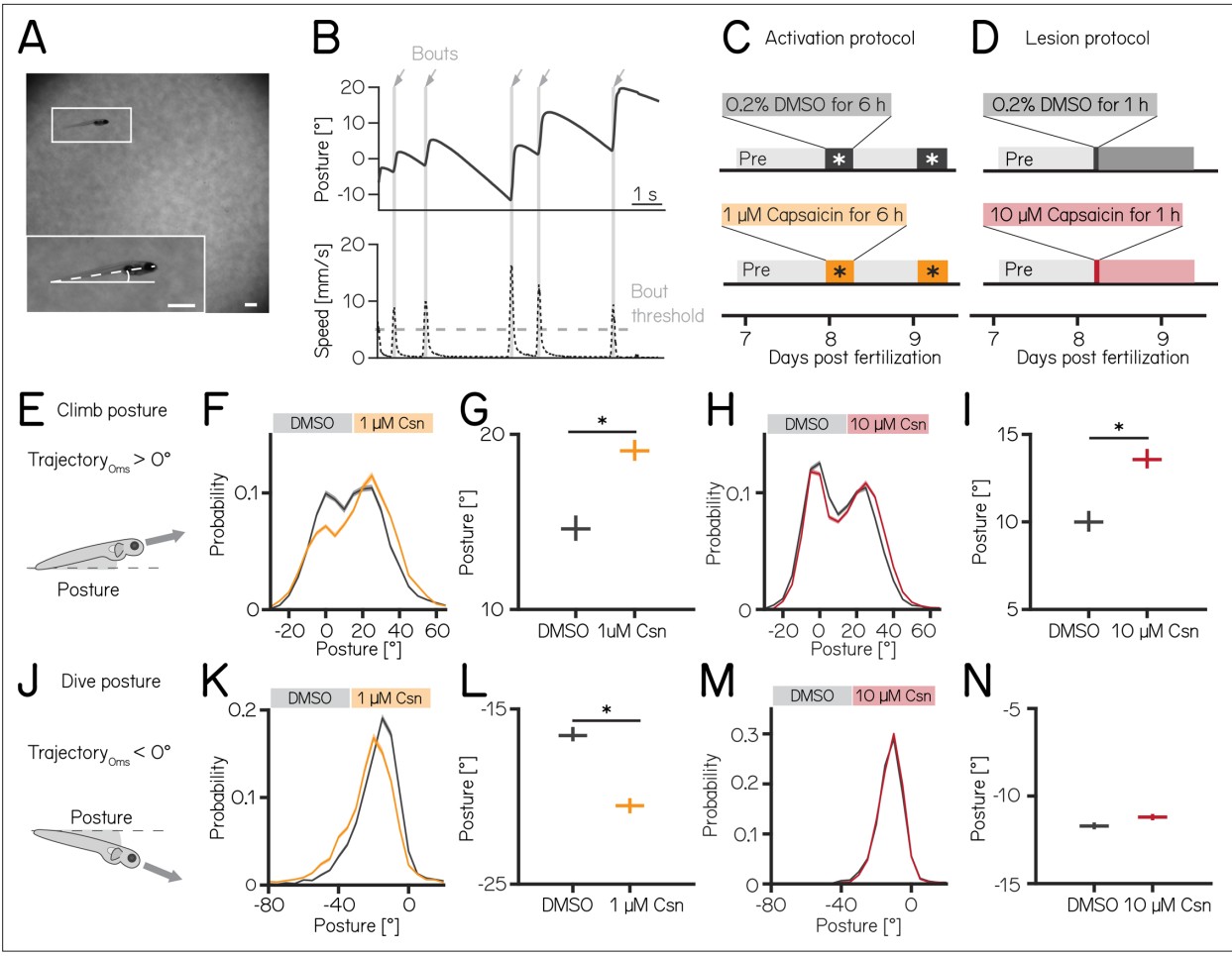

**Figure 2.** Both chemogenetic activation and ablation of Purkinje cells modify median postural pitch angle. (**A**) Sample image of a freely swimming zebrafish larva imaged from the side. Inset shows the larva at higher magnification view and its pitch, defined as the angle between the horizon (straight line) and the long axis of the body (dashed line). Scale bars 1 mm. (**B**) Pitch angle (posture, top) and speed (bottom) as a function of time for one recorded epoch. Individual swim bouts (speed > 5 mm/s threshold) are highlighted in grey (arrows). (**C**) Timecourse for activation experiments between 7 and 9 dpf. Larvae received 1 μM of capsaicin in 0.2% DMSO twice on days 8&9 for 6 hr each. (**D**) Timecourse for lesion experiments; larvae received a single dose of 10 μmM capsaicin in 0.2% DMSO for 1 hr on day 8. (**E**) Climbs are defined as a bout where the trajectory at peak speed took the fish nose-up (>0°). (**F**) Probability distribution of climb postures for control (black) and 1 μM capsaicin-treated larvae (yellow). Data is shown as median and inter-quartile range. (**G**) Average climb posture of control and activated larvae (8 repeats/149 control fish; 8 repeats/155 1 μM capsaicin-treated fish; climb postures: 14.7° [14.0–15.4°] vs. 19.0° [18.5–19.7°], p-value < 0.001, effect size: 29%, Wilcoxon rank sum test). (**H**) Probability distribution of climb postures for control (black) and 10 μM capsaicin-treated larvae (red). Data is shown as median and inter-quartile range. (**I**) Average climb posture of control and lesioned larvae (14 repeats/110 control fish; 14 repeats/120 10 μM capsaicin-treated fish; climb postures: 10.0° [9.5–10.7°] vs. 13.6° [13.1–14.3°], p-value < 0.001, effect size: 36%, Wilcoxon rank sum test). (**J–N**) Same as **E-I**, but for dive bouts (trajectory that took the fish in the nose-down direction). (**L**) Average dive posture of control and activated larvae (8 repeats/149 control fish; 8 repeats/155 1 μM capsaicin-treated fish; dive postures: –16.6° [-16.9 to –16.1°] vs. –20.5° [-20.9 to –20.1°], p-value < 0.001, effect size = 24%, Wilcoxon rank sum test). (**N**) Average dive posture of control and lesioned larvae (14 repeats/110 control fish; 14 repeats/120 10 μmM capsaicin-treated fish; dive postures: –11.7° [-11.9 to –11.5°] vs. –11.2° [-11.4 to –11.0°], p-value = 0.002, effect size = –4%, Wilcoxon rank sum test). Unless otherwise indicated data are shown as median with 95% confidence interval, * indicates p-value < 0.05 and effect size ≥15%.

The online version of this article includes the following figure supplement(s) for figure 2:

**Figure supplement 1.** Swim kinematics are not affected by 1 μM capsaicin treatment.

design yields data with large numbers of observations. To ensure a focus on only the most meaningful differences, we adopted two stringent criteria for significance: p-values <0.05, and an effect size of ≥15%. All p-values and effect sizes are reported in *Appendix 1—tables 1–5*.

We used the timing and capsaicin concentrations we had previously validated (*Figure 1*) to design two behavioral paradigms: one to activate and one to ablate cerebellar Purkinje cells. Experiments

were done from 7 to 9 dpf, and began with a single day without perturbations. Activation was then achieved by exposing *Tg(aldoca:TRPV1-tagRFP); Tg(elavl3:h2B-GCaMP6f)* fish to two 6 hr periods of 1 μM capsaicin while they swam freely in the dark (*Figure 2C*). Alternatively, Purkinje cells were ablated by exposing *Tg(aldoca:TRPV1-tagRFP)* fish to 10 μM of capsaicin for 1 hr (*Figure 2D*). All fish were screened before experiments for comparable levels of tagRFP fluorescence and control and experimental groups were randomly selected. A single experimental repeat consisted of 1–3 apparatus run in parallel with fish from a single clutch of embryos (i.e. siblings). To maintain consistency with genotypes used for validation, the activation and ablation experiments had different backgrounds (i.e. the presence/absence of the *elavl3:h2B-GCaMP6f* allele). Because of background variation (*Zhu et al., 2023*), all comparisons were restricted to control vs. experimental groups *within* an experimental paradigm over the same time period. We focused our analysis on the pitch axis as the current version of the SAMPL apparatus (*Zhu et al., 2023*) is not optimized for quantification of roll axis behavior. Across our datasets (*Appendix 1—tables 1–3*), we did not observe meaningful differences between the control and experimental groups in the pre-manipulation period. To avoid adding noise to our estimates of effect size, we therefore report comparisons between control and experimental groups after perturbation. We did not observe global consequences for swimming: swim speed, swim frequency, and bout duration were unaffected during Purkinje cell activation (*Figure 2—figure supplement 1A and B*) or after Purkinje cell lesion. Similarly, raw bout numbers were not different between the control and activation (median [inter-quartile range] 1256 [571–1454] bouts vs. 656 [444–1485] bouts; p-value 0.46) or lesion groups (2004 [1507–2471] bouts vs. 1913 [1556–2416] bouts; p-value 0.84, *Appendix 1—tables 1 and 2*).

Climbing postures were perturbed after both activation and ablation of Purkinje cells. During activation, fish adopted more nose-up postures before climb bouts. We observed a shift towards more positive values across the distribution of postures before fish initiated a climb bout (*Figure 2F*). The average climb posture of fish during depolarization was 29% higher than in control fish (*Figure 2G*, median [95% confidence interval]: 14.7° [14.0–15.4°] vs. 19.0° [18.5–19.7°], p-value < 0.001, effect size: 29%). Similarly, after Purkinje cell lesion, the average climb bout postural angle increased 36% relative to controls (*Figure 2H and I*, 10.0° [9.5–10.7°] vs. 13.6° [13.1–14.3°], p-value < 0.001, effect size: 36%); this increase replicates earlier findings that used a genetically encoded photosensitizer to ablate Purkinje cells (*Ehrlich and Schoppik, 2019*).

We observed an unexpected decrease in the climb bout postural angles for control fish in the post-lesion period (from 18.0° [17.6–18.4°] to 10.0° [9.5–10.7°], *Appendix 1—table 2*). We do not have an explanation for this particular change and we have confirmed that it is not due to a single outlier experiment (detected outliers: 0/15 pre-lesion control experiments; 0/14 post-lesion control experiments). Notably, if we assess the effect of adding 10 μM capsaicin by comparing the magnitude of the relative difference between pre- and post-lesion periods, normalized to the pre-lesion period, we still see a significant difference (control vs. lesion: –46% vs -26%). We conclude that, even when accounting for observed changes between control fish at 7 vs 8 dpf, Purkinje cell ablation modifies climb postures.

Dive bout postures were similarly perturbed after activation, but not ablation of Purkinje cells. Fish adopted more nose-down postural angles before dive bouts with a leftward shift of the distribution of postures before dive bouts (*Figure 2K*). Average dive bout posture was 24% more negative than in control fish (*Figure 2L*, median [95% confidence interval]: –16.6° [-16.9 to –16.1°] vs. –20.5° [-20.9 to –20.1°], p-value < 0.001, effect size = 24%). Purkinje cell lesions at 7 dpf did not shift the average posture for dive bouts (*Figure 2M and N* -11.7° [-11.9 to –11.5°] vs. –11.2° [-11.4 to –11.0°], p-value = 0.002, effect size = –4%).

We interpret these data as evidence that Purkinje cell activity is crucial to ensure that posture during climbs and dives is maintained within a normal range.

## Loss of Purkinje cells in older fish results in more global deficits to posture

Over the first two weeks of life, larval zebrafish morphology and postural control strategies develop considerably (*Ehrlich and Schoppik, 2017a*). These changes are matched by similarly pronounced cerebellar growth (*Hamling et al., 2015*; *Figure 3A and B*). We observed that the number of Purkinje cells labeled in *Tg(aldoca:TRPV1-tagRFP)* roughly doubled between 7 and 14 dpf (*Figure 3C*, median

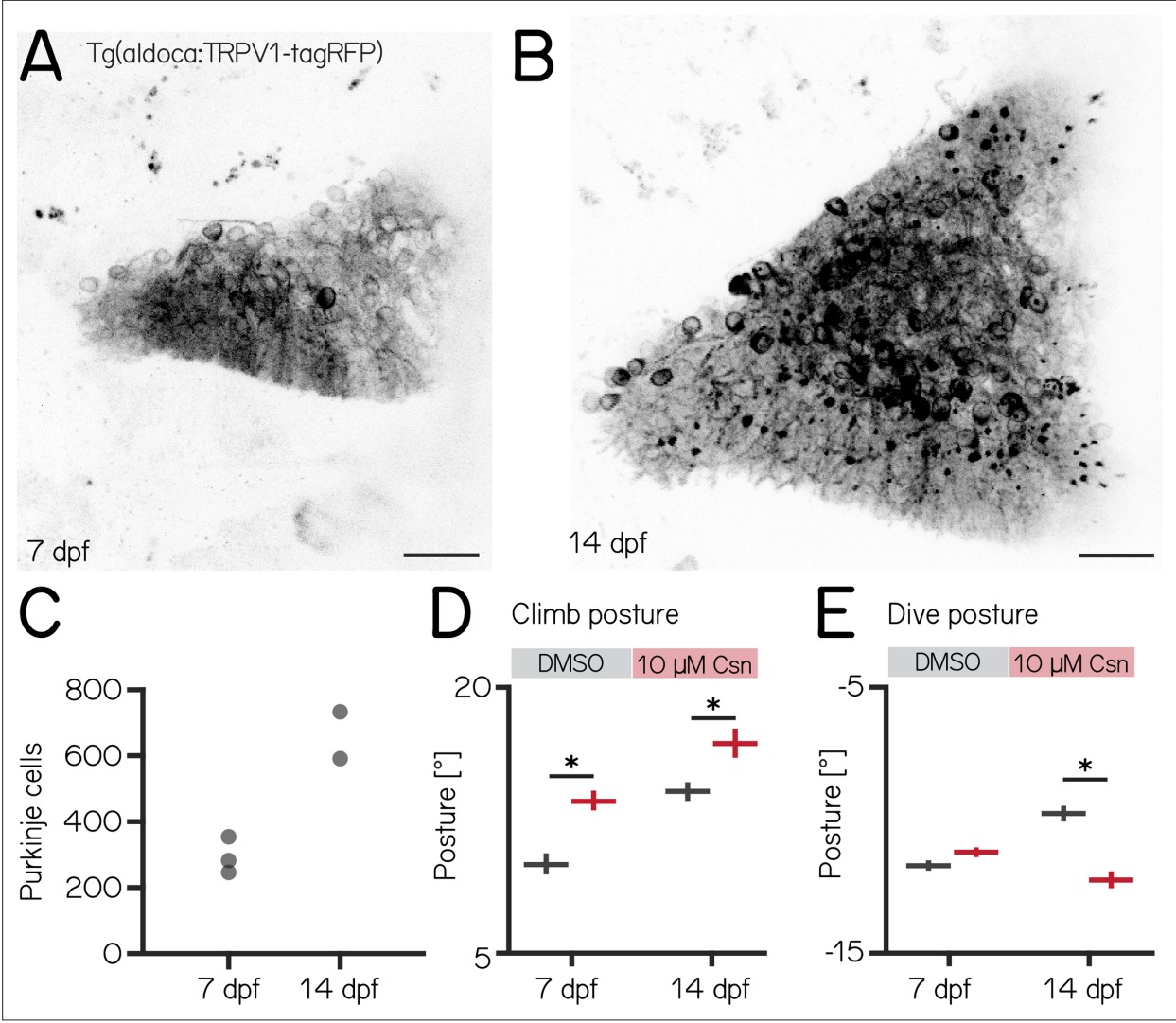

**Figure 3.** Changes to median postural pitch angle after chemogenetic ablation of Purkinje cells are more pronounced in older fish. (**A**) Confocal image of Purkinje cells in the cerebellum of a 7 dpf *Tg(aldoca:TRPV1-tagRFP)* larvae. Scale bar: 25 μm. (**B**) Confocal image of Purkinje cells in the cerebellum of a 14 dpf *Tg(aldoca:TRPV1-tagRFP)* larvae. Scale bar: 25 μm. (**C**) Increase in Purkinje cell numbers between 7 and 14 dpf. (**D**) Average climb bouts postures for 7 dpf control and lesion larvae (left) and 14 dpf control and lesion larvae (right). (14 dpf lesion: 7 repeats/48 control fish; 7 repeats/44 10 μM capsaicin-treated fish; climb postures: 14.3° [13.8–14.8°] vs. 17.1° [16.2–17.8°]; p-value < 0.001; effect size: 20%, Wilcoxon rank sum test). (**E**) Average dive bouts postures for 7 dpf control and lesion larvae (left) and 14 dpf control and lesion larvae (right). (14 dpf lesion: 7 repeats/48 control fish; 7 repeats/44 10 μM capsaicin-treated fish; dive postures: –9.8° [-10.1 to –9.5°] vs. –12.3° [-12.6 to –11.9°]; p-value < 0.001; effect size: 26%, Wilcoxon rank sum test). all data are shown as median with 95% confidence interval, * indicates p-value < 0.05 and effect size ≥15%.

The online version of this article includes the following figure supplement(s) for figure 3:

**Figure supplement 1.** Purkinje cell lesion at 14dpf affects the distribution of postural angles for climb and dive bouts.

[inter-quartile range] 7 dpf: 282 [255 – 336]; 14 dpf: 662 [591 – 733]). The increase in cell numbers is also evidence that the *aldoca* promoter continued to drive expression at later stages, allowing us to perform comparative experiments.

Similar to lesions at 7 dpf, we did not observe any differences in swim speed, frequency or bout duration (*Appendix 1—table 3*). At 14 dpf, the effects of Purkinje cell lesions on postural angles were more widespread than at 7 dpf and also affected dive postures. We repeated our previous ablation experiments (*Figure 2D*) between 14–16 dpf, and analyzed climb (Figures 3 and 8) and dive bouts (*Figure 3E*, *Figure 3—figure supplement 1B*). Loss of Purkinje cells created more widespread behavioral deficits. Specifically, climb bout posture was increased by 20% after Purkinje cell lesion (median [95% confidence interval]: 14.3° [13.8–14.8°] vs. 17.1° [16.2–17.8°]; p-value < 0.001; effect size: 20%).

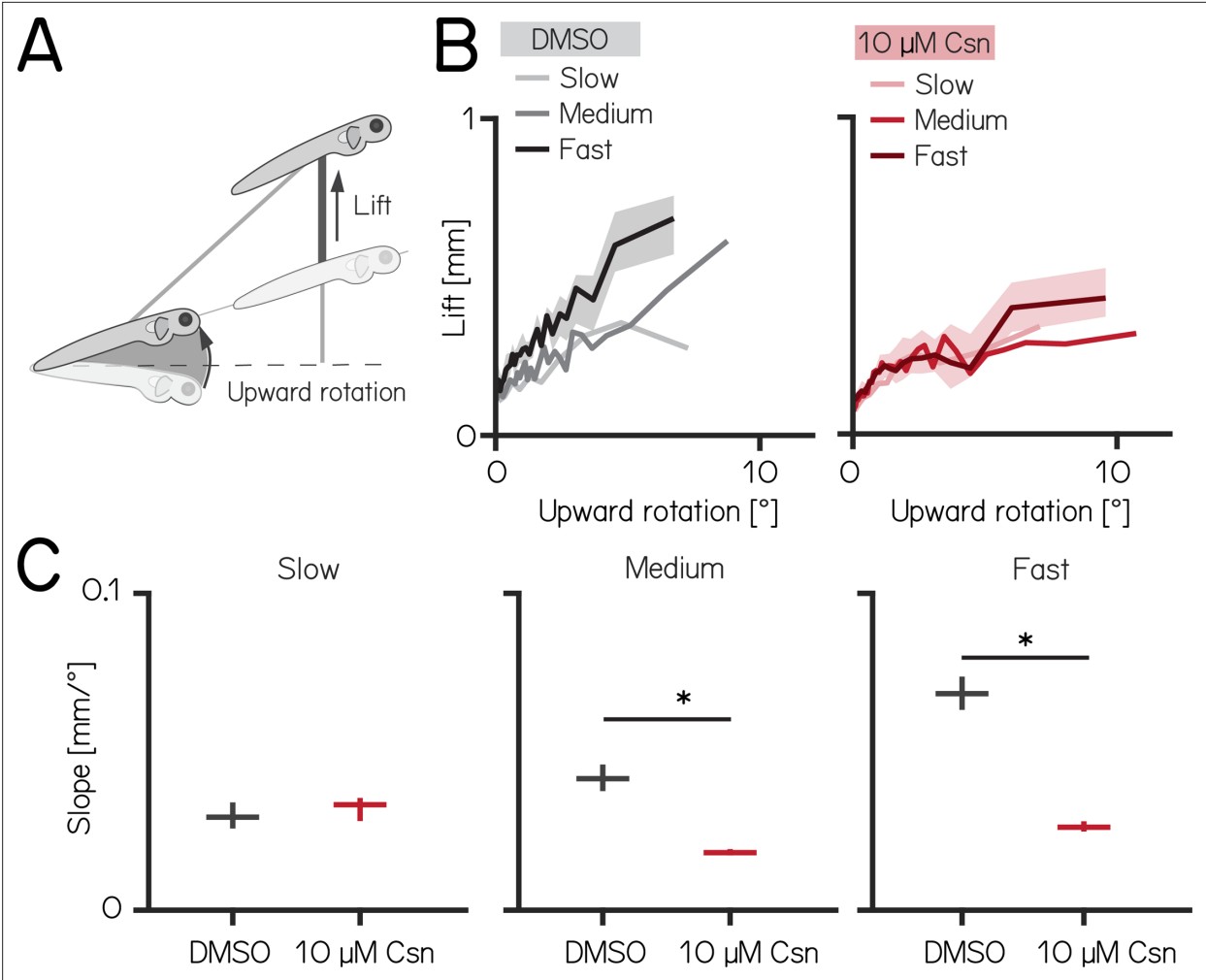

**Figure 4.** Chemogenetic ablation of Purkinje cells disrupts fin-body coordination in a speed-dependent manner. (**A**) Larval zebrafish use two independent effectors (trunk and body) to climb. The contribution of each effector can be dissociated by the observed kinematics: changes to the angle of the trunk predict a trajectory for a particular bout (upward rotation). The actual position of the fish in depth at the end of the bout reveals the lift generated by the fins. A detailed kinematic examination of climbing, including fin ablations, is detailed in *Ehrlich and Schoppik, 2019*. (**B**) Coordination of fin and trunk engagement plotted as upward rotation against lift. Positive slopes reveal that larger rotations are coupled to greater fin engagement and greater changes in depth. The slope of this relationship becomes steeper for bouts with greater translational speed. Bouts from control (grey,left) and 10 μM capsaicin-treated larvae (red,right) are plotted at different swim speeds, shaded areas indicate 95% confidence interval of the median of the fast swim speeds. (**C**) Average slopes of lift/rotation curves for control and 10 μM capsaicin-treated larvae at different swim speeds. (8 repeats/15 control fish; 8 repeats/18 10 μM capsaicin-treated fish); slow: p=0.341; medium: p<0.001; fast: p<0.001. Data are plotted as median with inter-quartile range. * indicates p < 0.05 and effect size ≥15%.

The online version of this article includes the following figure supplement(s) for figure 4:

**Figure supplement 1.** Fin engagement is speed dependent.

At 14 dpf we also observed an effect on dive bout postures. After lesion dive bouts postures were 26% more negative (–9.8° [-10.1 to –9.5°] vs. –12.3° [-12.6 to –11.9°]; p-value < 0.001; effect size: 26%).

We conclude that, consistent with morphological growth, Purkinje cells of the cerebellum play a broader role in postural control at 14 dpf than at younger ages.

## Purkinje cells regulate speed-dependent fin engagement

To climb, larval zebrafish coordinate fin movements that generate lift with axial rotations that direct thrust (*Figure 4A*). The greater the axial rotation, the stronger the lift-producing fin movements; this relationship increases as larvae develop (*Ehrlich and Schoppik, 2019*). Our previous work suggested that Purkinje cells were necessary for such fin-body coordination (*Ehrlich and Schoppik, 2019*). Here,

we observed that fin engagement is speed-dependent, with faster bouts producing greater lift for a given axial rotation (*Figure 4B*, left, Spearman's correlation coefficient: 0.2193; p = <0.001; of lift/rotation ratio [mm/deg] versus speed [mm/s] *Figure 4—figure supplement 1A*).

After Purkinje cell ablation, 14 dpf fish produced less lift than expected when they swam fast. We divided swim bouts into three different bins according to their peak speed (slow: 5–7.5 mm/s; medium: 7.5–15 mm/s; fast >15 mm/s) for both control and fish treated with 10 µM capsaicin. We parameterized the relationship between upward rotation and lift by fitting a line to swim bouts for each speed. After capsaicin exposure, the slopes of the medium and fast speed bins were significantly lower (*Figure 4C*, *Figure 4—figure supplement 1*), reflecting a loss of speed-dependent modulation (median [95% confidence interval]: slope slow: 0.029 [0.028–0.031 mm/°] vs. 0.033 [0.032–0.034 mm/°], p-value = 0.341, effect size: 6%; slope medium: 0.041 [0.040–0.044 mm/°] vs. 0.018 [0.018–0.019 mm/°], p-value <0.001, effect size: –34%; slope fast: 0.068 [0.067–0.071 mm/°] vs. 0.026 [0.026–0.027 mm/°], p-value <0.001, effect size: –62%;). The correlation between speed and fin-lift / rotation ratio is reduced after capsaicin exposure (*Figure 4—figure supplement 1*; Spearman correlation coefficient: control 0.2193; 10 µM capsaicin: 0.0397; Z-test after z-transformation: p < 0.001) When analyzing fin lift and upward rotation across the three speed bins separately, we only observed significant differences for fast swim bouts, specifically showing a reduction in fin lift and an increase in upward rotation *Appendix 1—table 4*.

Next, to determine if lift was fin-dependent, we amputated the fins (*Ehrlich and Schoppik, 2019*, *Zhu et al., 2023*) and repeated our experiments. A detailed explanation of how fin amputation affects swim kinematics can be found here (*Ehrlich and Schoppik, 2019*, *Zhu et al., 2023*). We observed a

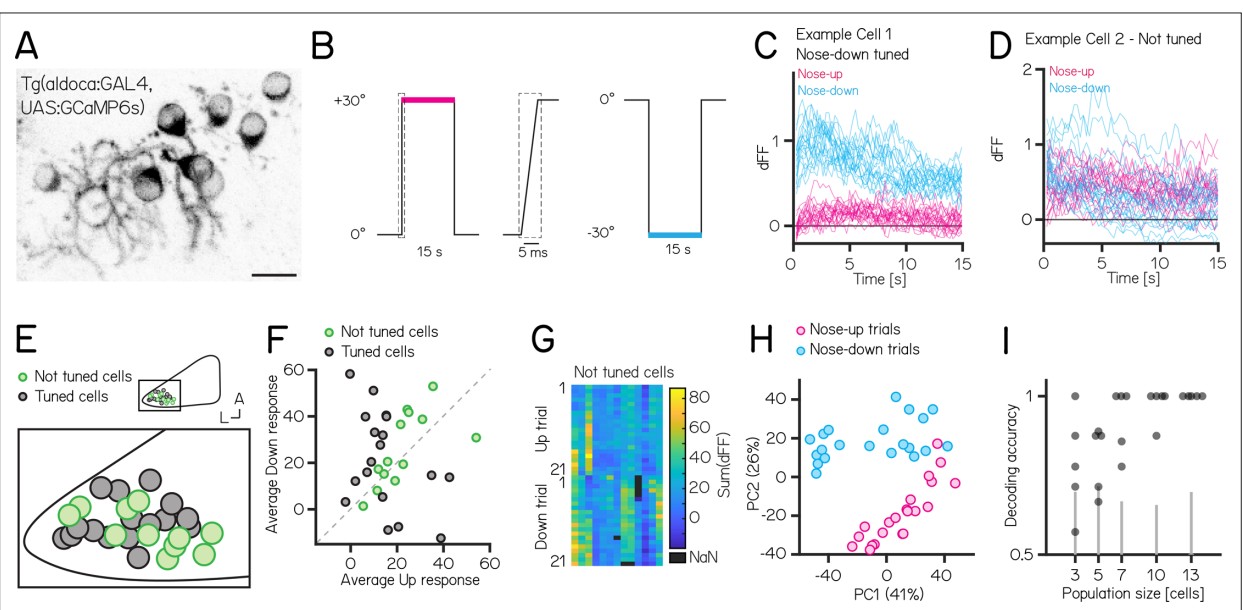

**Figure 5.** Activity in larval zebrafish Purkinje cells can differentiate nose-up from nose-down pitch both individually and collectively. (**A**) Two-photon image of Purkinje cell somata expressing a calcium indicator in the *Tg(aldoca:GAL4);Tg(UAS:GCaMP6s)* line. Scale bar 10 µm. (**B**) Pitch tilt stimuli consisted of rapid galvanometer steps for 15 s in the nose up (+30°, pink) and nose-down (–30°, blue) direction. Inset in dotted rectangle shows the near-instantaneous timecourse of the step. (**C**) Example responses (n=42) from a single Purkinje cell sensitive to nose-down pitch (blue) but not nose-up (pink). (**D**) Example responses (n=42) from a single Purkinje cell without directional selectivity. (**E**) Superimposed positions of Purkinje cell somata within a single cerebellar hemisphere; no obvious topography separates tuned (black, n=16) and untuned (green, n=11 —directionality index— < 0.35) cells. (**F**) Averaged integrated response (dFF) for individual cells over the 15 s stimulus plotted for nose-up vs. nose-down stimuli, colored by tuned (black) and untuned (green). (**G**) Heatmap of integrated response (dFF) for 13 untuned neurons on 21 up/down tilts. (**H**) Principal component analysis of integrated responses for untuned neurons for each of 21 up (pink) and 21 down (blue) trials. (Percentage of variance explained) (**I**) Performance of a support vector machine for binary classification of up/down tilt using integrated responses from increasing numbers of untuned neurons. Dots are different sets of neurons, gray lines shows the spread of performance from shuffled up/down identity (median [interquartile range] accuracy: 3/5/7/10/13 cells: 0.78 [0.68–0.91] / 0.88 [0.70–0.88] / 1 [0.84–1] / 1 [0.97–1] / 1 [1 – 1]).

The online version of this article includes the following figure supplement(s) for figure 5:

**Figure supplement 1.** Purkinje cell tuning direction shifts across development, population coding strength remains stable.

near total loss of lift at all speeds; regardless of the speed bin, the slope of the relationship between upward rotation and lift was indistinguishable from zero (slope slow: 0.036 [0.035–0.037 mm/°] vs. –0.005 [-0.005 to -0.004 mm/°], p-value <0.00.1; effect size: –60%; slope medium: 0.055 [0.054–0.056] mm/° vs. –0.005 [-0.005 to -0.005 mm/°], p-value <0.001; effect size: –88%; slope fast: 0.068 [0.067–0.069 mm/°] vs. 0.013 [0.013–0.013 mm/°], p-value <0.001; effect size: –81%). Finally, we examined fin-body coordination in our 7 dpf activation and ablation datasets. In contrast to older larvae, we observed no meaningful changes after activation of Purkinje cells at 7 dpf. For Purkinje cell lesions at 7 dpf we found only the fin body coordination at fast bouts to be affected *Appendix 1—tables 1 and 2*.

Our data show that loss of Purkinje cells disrupts the speed-dependent increase in fin-mediated lift in older, and to a lesser degree in younger fish. We interpret this finding as evidence that Purkinje cells are indispensable for normal coordination of the fins and body.

## Purkinje cells encode pitch direction at both individual and population levels

Our experiments establish that manipulations of Purkinje cells interfere with balance in the pitch axis. We therefore hypothesized that Purkinje cell activity would be modulated by nose-up/nose-down body tilts. We used Tilt In Place Microscopy (TIPM) (*Zhu et al., 2023*) to measure the response of individual Purkinje cells (*Figure 5*) to rapid pitch tilts. Briefly, fish are mounted on a mirror galvanometer and rapidly rotated to eccentric angles (*Figure 5B*, ±30°nose-up/nose-down).

We used *Tg(aldoca:GAL4);Tg(UAS:GCaMP6s)* to label Purkinje cells in the lateral parts of the cerebellum thought to receive vestibular input (*Bae et al., 2009*, *Matsui et al., 2014*, *Knogler et al., 2019*). To facilitate identification of the same cells from volumes imaged at both ±30°, we used doubly mono-allelic fish and screened for sparse expression of Purkinje cells. In total, we imaged 43 Purkinje cells from 8 fish. Of those, 31 cells could reliably identified at ±30° and were included in the analysis.

We calculated a directionality index (DI) for all cells and categorized cells as either tuned (—DI— >0.35) or as untuned (—DI— < 0.35). Some cells showed, on average, higher responses to one direction (*Figure 5F*), but due to highly variable responses (median [inter-quartile range] Up response range: 23.3 [18.3–29.5]; Down response range: 43 [23.7–49.2]), they did not exhibit consistent tuning as determined by the directionality index. Individual Purkinje cells showed either directionally-tuned (*Figure 5C*, n=18) or untuned (*Figure 5D*, n=13) patterns of responses. Tuned cells were distributed throughout the lateral cerebellum (*Figure 5E*), and showed a slight preference for nose-down stimuli (12 vs 6, *Figure 5F*). We did not observe any systematic differences in the response properties across each experiment from untuned cells (*Figure 5G*).

While untuned cells did not show overt directional preferences, pooling their responses allowed decoding of stimulus direction. We applied principal component analysis (PCA) to visualize the data and assess whether the trial types even for untuned cells exhibited distinct separation based on their trial identity (nose-up or nose-down). PCA of the integral of the full responses on each trial from untuned neurons showed near-complete segregation of trial types (*Figure 5H*). To assay whether there was indeed directional information we trained a decoder (support vector machine) and tested its accuracy on pseudo-populations of untuned cells of different sizes ranging from 3 to 13 cells (*Figure 5I*). We focused only on untuned cells, as including even a single tuned cell for the population coding will lead to excellent results. Training and test trials were different to avoid over-fitting. Pseudo-populations with more than three cells achieved accurate decoding well above chance levels (determined by shuffling trial identity)(median [inter-quartile range] accuracy: 3/5/7/10/13 cells: 0.78 [0.68–0.91] / 0.88 [0.70–0.88] / 1 [0.84–1] / 1 [0.97–1] / 1 [1 – 1]). To ensure that the choice of cutoff for the directionality index (0.35) does not bias the decoding results, we also calculated the decoding accuracy using the 3, 5, and 7 least directionally tuned cells based on their directionality index. We again found that more than 3 cells achieve accurate decoding above chance level (3/5/7 least tuned cells: 0.67 [0.64–0.78] / 0.86 [0.73–0.88] / 0.88 [0.86–0.92]).

Older larvae showed additional changes to dive postures after Purkinje cell lesions. We therefore tested if: (1) Purkinje cells in older larvae exhibited differences in the numbers or direction of tuned cells or (2) if population-level decoding accuracy changed. We performed longitudinal TIPM, sampling from zebrafish larvae at 7 and 14 dpf. To improve throughput, we recorded the responses upon return from ±19° stimuli (*Figure 5—figure supplement 1*; 7 dpf: 138/11 cells/fish; 14 dpf: 90/7 cells/fish; of those 23/3 cells/fish were imaged at both timepoints); previous work established that responses upon

return to baseline are highly correlated with the response at the eccentric position (*Hamling et al., 2023b*). We observed increased fluorescence relative to baseline values in 7 dpf and 14 dpf Purkinje cells upon return from ±19° steps (*Figure 5—figure supplement 1*). To analyze directional tuning we compared the maximum fluorescence in the first second after return to baseline. The relative number of tuned cells per fish was comparable between 7 and 14 dpf larvae (*Figure 5—figure supplement 1*; median [inter-quartile range] at 7 dpf: 7 [6–34%]; 14 dpf: 8 [2–19%]; p-value = 0.7763). While most cells were not directionally selective, the preferred direction of tuned cells was different at 7 and 14 dpf: at 7 dpf more Purkinje cells were nose-down tuned (2/31 up/down) but at 14 dpf more cells were nose-up tuned (11/3 up/down *Figure 5—figure supplement 1*; Fisher's exact test, p-value < 0.001).

We next assayed accuracy of directional encoding of simultaneously recorded untuned cells. We performed PCA for untuned cells at 7 (*Figure 5—figure supplement 1*) and 14 dpf (*Figure 5—figure supplement 1*) and tested decoding accuracy on the untuned cells of individual fish at 7 and 14 dpf. We did not observe differences in decoding accuracy between 7 and 14 dpf larvae (*Figure 5—figure supplement 1*; median [inter-quartile range] 7 dpf: 0.68 [0.63–0.83]; 14 dpf: 0.73 [0.65–0.79]; p-value = 0.9468). We conclude that cerebellar Purkinje cells can encode pitch direction both at the single neuron and population levels with similar encoding accuracy in young and older larvae.

## Discussion

We used a validated chemogenetic tool to investigate the role of cerebellar Purkinje cells in postural behavior as larval zebrafish swam freely in depth. Activation of Purkinje cells could induce changes in pitch axis (nose-up/nose-down) posture. Purkinje cell ablation changed posture, with broader effects in older larvae. Ablation disrupted fin-body coordination responsible for proper climbing. Finally, we could reliably decode pitch-tilt direction from patterns of Purkinje cell activity. We did not observe developmental changes in population coding of direction but found a shift in the tuning direction of Purkinje cells. Taken together our results establish a role for the cerebellum in postural control even during the earliest stages of larval zebrafish development. Our work establishes a new method that combines bidirectional manipulation of cerebellar output and quantitative high-throughput measures of unconstrained posture and locomotion.

### Contributions of Purkinje cells to posture

While activation and ablation manipulations both produced biologically meaningful behavior changes, the two experiments were run with different genetic backgrounds and on different generations of the SAMPL apparatus. Consequentially, our ability to define precisely what role Purkinje cells play in balance behaviors in larval zebrafish is limited. Activation experiments are particularly laborious as they require thorough pre-screening to ensure adequate brightness levels to achieve sufficient depolarization without excitotoxicity. To maintain consistency in depolarization effects, experiments involving different genetic backgrounds or developmental stages would require additional calcium imaging to confirm that 1 µM capsaicin elicits comparable responses. Hence, we restricted our experiments to 7 days post-fertilization (dpf) and did not extend the study to other developmental time points. Given that the primary purpose of this series of experiments was to establish TRPV1-mediated manipulation of Purkinje cells as a means to investigate postural control, it is beyond the scope of the work to repeat the experiments. Nonetheless, we consider the findings individually below in the context of prior work.

Purkinje cell ablations modifies postural stability. Importantly, the differences we observed were more widespread in older larvae, underscoring the developmental importance of Purkinje cells for balance. Purkinje cell output is inhibitory (*Ito et al., 1964*, *Ito et al., 1970*), Purkinje cells in the lateral cerebellum project to vestibular nuclei (*Matsui et al., 2014*, *Knogler et al., 2019*), and Purkinje cells are tonically active (*Cerminara and Rawson, 2004*, *Jadhav et al., 2023*). We propose that the net effect of Purkinje cell loss would be disinhibition of target nuclei responsible for encoding posture and parameterizing corrective pitch-axis behaviors. While the precise nature of the transformation between larval zebrafish pitch and posture control kinematics is not yet known, loss of cerebellar-targeted nuclei can disrupt postural behaviors (*Hamling et al., 2021*, *Ehrlich and Schoppik, 2019*).

The effects of ablations became more widespread in older larvae. During early development, larval zebrafish grow in volume by roughly an order of magnitude and shift their postural control

strategies to better climb/dive as they navigate in depth (*Ehrlich and Schoppik, 2017a*, *Ehrlich and Schoppik, 2019*). Unlike climb bouts, changes to postural stability during dives only emerge at 14 dpf. As activation of Purkinje cells produced meaningful changes during dives at 7 dpf, we infer that the delayed emergence of ablation effects does not reflect incomplete integration of Purkinje cells into dive-control circuits. Notably, the basal posture during dive bouts decreases in older control fish (*Figure 3E*) — ablation shifts the posture comparable to its younger state. Future work with our system enables testing of the hypothesis that Purkinje cell output plays a role in setting the postures older fish adopt during dives.

Purkinje cell activation also modifies postural stability. Intriguingly, activation broadened the distribution of observed postures in the same way as ablation. Our imaging assay established that 1 µM of capsaicin would stochastically activate subsets of Purkinje cells. This stochasticity could reflect normal fluctuations in basal levels of activity, or it could arise from cells going in and out of depolarization block (*Mattis et al., 2011*). Synchronized/precisely timed Purkinje cell output is thought to shape movements (*Heck et al., 2007*, *Person and Raman, 2011*, *Person and Raman, 2012*, *Han et al., 2018*, *Payne et al., 2019*, *Nashef et al., 2023*), although perhaps not for all behaviors (*Herzfeld et al., 2023*). Our imaging suggests that the set of Purkinje cells activated at any one moment in time is limited and random. We therefore propose that the net effect of 1 µM of capsaicin is ultimately disruptive to Purkinje cell synchrony, and thus likely disruptive. Future work could test this hypothesis by intracellular recording from cerebello-recipient populations like the vestibular nuclei (*Liu et al., 2020*, *Hamling et al., 2023a*).

Previously, we reported that larval zebrafish coordinate their fins and trunk to climb effectively *Ehrlich and Schoppik, 2019*. The relationship between trunk-mediated changes to trajectory (upward rotation) and fin-mediated lift depends on locomotor speed. Here, we observed that after Purkinje cell loss, speed-dependent increases in lift with greater trunk rotation are disrupted (*Figure 4C*, *Figure 4—figure supplement 1A and B*). In the fastest speed bin, we observed an increase in upward rotation and a decrease in average fin lift. In contrast, the medium-speed bin showed no significant changes in average fin lift or upward rotation, yet already displayed coordination deficits. Based on these observations, we argue that Purkinje cell lesions primarily affect coordination, rather than simply reducing one specific parameter such as lift or rotation. As we did not observe any change to locomotor speed after ablation (*Appendix 1—tables 1–3*), we infer that Purkinje cell loss disrupts speed-dependent coordination for climbing. These results extend our original report where a lower throughput method (photoablation) suggested that Purkinje cell loss impacted the fin-trunk relationship (*Ehrlich and Schoppik, 2019*). In larval zebrafish, the neuronal substrates for axial speed control (*Severi et al., 2014*; *Wang and McLean, 2014*; *Thiele et al., 2014*; *Kishore et al., 2020*; *D'Elia et al., 2023*; *Carbo-Tano et al., 2023*) and fin engagement (*Uemura et al., 2020*) are known. The potential for whole-brain imaging in larval zebrafish (*Lin et al., 2020*), particularly with high-speed voltage indicators (*Böhm et al., 2022*) and cutting-edge modeling approaches (*Costabile et al., 2023*), stands to reveal how Purkinje cell activity comes to coordinate body and fin movements. Importantly, since our behavioral data suggest that Purkinje cell activity impacts fin-trunk coordination more strongly in older larvae, longitudinal approaches will be key to understanding the developmental changes to cerebellar signaling that underlie effective coordination of trunk and limbs.

## Encoding strategies for body tilt stimuli

Purkinje cell activity reflects both sensory and motor inputs. One limitation of TIPM is that larvae are immobilized in agarose during tilts. Consequentially, our measurements of Purkinje cell activity are artificially constrained. Nonetheless, a subset of Purkinje cells were unambiguously direction-selective, and a simple decoder could use the activity of non-selective cells to differentiate tilt direction. We infer that vestibular information directly related to pitch axis posture is represented by the Purkinje cell population targeted in our ablation/activation experiments, consistent with broader imaging of cerebellar responses to body tilt (*Migault et al., 2018*, *Favre-Bulle et al., 2018*). Similar to the behavior results, we observed an asymmetry in the tuning direction of Purkinje cells at 7 dpf, with more cells being tuned to the nose-down direction. This asymmetry shifted between 7 and 14 dpf, suggesting developmental changes in how navigation in the pitch axis is processed in the cerebellum. These changes underscore the importance of longitudinal measurements of Purkinje cell activity across early development to understand emergent control of posture.

The ability to decode tilt direction from the collective activity of 'untuned' Purkinje cells suggests a role for population coding. Such mechanisms have been proposed for head/body motion (*Zobeiri and Cullen, 2022*) and eye movements (*Sedaghat-Nejad et al., 2022*, *Herzfeld et al., 2023*) in the primate cerebellum. Population coding requires that multiple Purkinje cells converge onto downstream targets, which is well-established in cerebellar target nuclei (*Person and Raman, 2011*, *Heck et al., 2013*). In larval zebrafish, Purkinje cells involved in locomotion converge on eurydendroid cells; electrophysiological recordings confirm a many-to-one convergence scheme that could similarly support population coding (*Harmon et al., 2017*). Vestibular-sensitive cells are located in the lateral cerebellum (*Migault et al., 2018*, *Favre-Bulle et al., 2018*), which projects to hindbrain regions that contain vestibular nuclei (*Hamling et al., 2015*). Comparing activity of vestibular nucleus neurons involved in tilt-driven behaviors (*Bianco et al., 2012*, *Liu et al., 2020*, *Sugioka et al., 2023*) before/after TRPV1-mediated ablation would speak to the collective contributions of Purkinje cells.

## TRPV1/capsaicin as a tool to study cerebellar contributions to behavior

Our use of TRPV1/capsaicin complements a modern suite of molecular tools to target cerebellar Purkinje cells (*Prestori et al., 2020*). In fish, different experiments have used opsins to excite / inhibit cerebellar Purkinje cells with exceptional temporal precision, establishing functional topography (*Matsui et al., 2014*) and an instructive role in learning (*Harmon et al., 2017*). TRPV1/capsaicin is a well-validated approach (*Chen et al., 2016*) that permits parametric (i.e. dose-dependent) activation/ablation with a single transgenic line. It does not require light, facilitating dissociation of vestibular from visual contributions without requiring genetically blind fish as in other studies using excitatory opsins (*Schoppik et al., 2017*). Additionally, our approach of using TRPV1/capsaicin for cell ablation offers multiple advantages compared to other lesion methods. For example, Killer Red requires extended exposure to high light intensities and mounting of single fish (*Ehrlich and Schoppik, 2019*). Nitroreductase ablation, while effective, requires extended pro-drug exposure, leading to slower and less precise results (*Curado et al., 2008*).

In contrast, TRPV1-mediated ablation is rapid and precise, with capsaicin triggering almost instantaneous cell death. This speed and simplicity make TRPV1 superior for experiments requiring quick, controlled ablation without the delays associated with other methods. Finally, chemogenetic approaches such as TRPV1/capsaicin permit prolonged experimentation in freely moving animals, allowing us to collect large kinematic datasets necessary to rigorously study posture and locomotion.

Considerable progress has been made in recent years using new tools (*Machado et al., 2015*, *Sheppard et al., 2022*, *Darmohray et al., 2019*, *Machado et al., 2020*, *Jaarsma et al., 2023*) and new perspectives (*De Zeeuw et al., 2021*) to understand the cerebellar contributions to sensorimotor (*Heiney et al., 2021*, *Gaffield et al., 2022*, *Verpeut et al., 2023*, *Soetedjo and Horwitz, 2023*) and non-sensorimotor behaviors (*Carta et al., 2019*, *Chen et al., 2022*, *Jackman et al., 2020*, *Zamudio et al., 2023*) in health and disease (*Chao et al., 2020*, *Chao et al., 2021*, *van der Heijden et al., 2023*). Underlying this considerable progress is an ever-improving ability to manipulate the cerebellum without compromising rigorous measures of behavior. Here — in support of similar goals — we validated a new chemogenetic approach (TRPV1/capsaicin-mediated activation and ablation) compatible with a high-throughput paradigm to measure behavior in freely swimming larval zebrafish (SAMPL). Our data uncover expected signatures of cerebellar contributions to posture and coordination, establishing the validity of our approach. Further, by comparing the impact of Purkinje cell ablation in time, we leverage the rapid maturation of the zebrafish to open a window into cerebellar control of posture and coordination across development. Our approach establishes a path forward for the larval zebrafish model to contribute to cerebellar mechanisms of postural control. The cerebellum emerged early in the evolution of vertebrates, when vertebrate life was underwater. Our work establishes a new tool to investigate ancient organizing principles of cerebellar function.

# Materials and methods
## Fish care

All procedures involving zebrafish larvae (*Danio rerio*) were approved by the Institutional Animal Care and Use Committee of New York University. Fertilized eggs were collected and maintained at 28.5 °C on a standard 14/10 hour light/dark cycle. Before 5 dpf, larvae were maintained at densities of 20–50

larvae per petri dish of 10 cm diameter, filled with 25–40 mL E3 with 0.5 ppm methylene blue. After 5 dpf, larvae were maintained at densities under 20 larvae per petri dish and fed cultured rotifers (Reed Mariculture) daily.

## Fish lines

To generate the *Tg(aldoca:TRPV1-TagRFP)* line, the 5-kbp aldolase Ca (aldoca) promoter (*Tanabe et al., 2010*) and a gene cassette that includes TRPV1-Tag1RFP cDNA, rabbit beta-globin intron, and the SV40 polyadenylation signal (pAS) in pT2-4xUAS:TRPV1-RFPT (*Chen et al., 2016*) were subcloned into the Tol2 vector pT2KDest-RfaF (*Nojima et al., 2010*) by the Gateway system (pT2K-aldoca-TRPV1-Tag1RFP-pAS). To establish stable transgenic lines, Tol2 plasmid and transposase mRNA (25 ng/ µl each) were injected into one-cell-stage embryos.

The resulting *Tg(aldoca:TRPV1-tagRFP)* stable line allowed us to express the mammalian capsaicin-sensitive cation channel TRPV1 and the red fluorophore tagRFP in cerebellar Purkinje cells. Before exposure to capsaicin, fish were screened to ensure similar levels of tagRFP expression. When screening for transgene expression, we selected fish with clearly visible expression that was not excessively bright. The same criteria were applied when screening fish for GCaMP imaging and behavior experiments. Approximately a quarter of the fish that had aldoca:TRPV1-tagRFP expression had suitable expression levels for the activation experiment.

We measured neuronal activity using a genetically encoded calcium indicator, *Tg(UAS:GCaMP6s)* (*Thiele et al., 2014*), driven by *Tg(aldoca:GAL4)* (*Takeuchi et al., 2015*), or the *Tg(elavl3:h2B-GCaMP6f)* line (*Dunn et al., 2016*).

## Confocal imaging of TRPV1-mediated activation / lesion

Images were collected using a Zeiss LSM800 confocal microscope using a 20x1.0 NA water immersion objective. Larvae were mounted in 2% low melting point agar (catalog #16520, Thermo Fisher Scientific) in a dorsal up position. Anatomical images were acquired from fish anesthetized with 0.2 mg/ml ethyl- 3-aminobenzoic acid ethyl ester (MESAB, catalog # E10521, Sigma-Aldrich). To activate TRPV1-expressing Purkinje cells, fish were treated with 1 µM capsaicin in 0.2% DMSO in E3. To lesion Purkinje cells, fish were exposed to 10 µM capsaicin in 0.2% DMSO in E3. Control fish were treated with 0.2% DMSO in E3. Agar was removed around the tip of the tail to facilitate drug delivery. Fish were mounted throughout functional imaging experiments and kept in temperature controlled incubators between timepoints. Time series images were acquired with a two-photon microscope (Thorlabs Bergamo equipped with a Mai Tai HP laser tuned to 920 nm) with a framerate of 7.9 fps. Images were analyzed in Fiji (*Schindelin et al., 2012*); ROIs were drawn on nuclei of randomly selected Purkinje cells, which were then re-identified at each time point. Fluorescence for each cell and time point was normalized to the pre-capsaicin value. Cells with a dF/F0 of > 2 were considered activated and analyzed at the pre capsaicin and + 6 hr timpoint. A Fisher's exact test was performed to determine significance of the activated cells.

To image the anatomy of Purkinje cells exposed to 10 µM of capsaicin across time, the cerebellum was imaged at 7 dpf from fish mounted as above. Fish were unmounted and kept in E3 medium until the next day (8 dpf). At 8 dpf, fish were placed in 0.2% DMSO in E3 (control) or 10 µM capsaicin in 0.2% DMSO in E3 for 40–60 min, and imaged again after 1 hr of recovery in E3 post-treatment. Fish from both groups were imaged again at 9 dpf. Confocal images were analyzed in Fiji and Purkinje cell somata were counted in both hemispheres of the cerebellum. A conservative approach was taken for cell counting, with inclusion of any structures still resembling cells, regardless of potential non-functionality or signs of degradation. Consequently, the counts are likely an underestimate of the actual percentage of cell loss.

## Zebrafish behavior recordings

All behavior was measured using the Scalable Apparatus for Measuring Posture and Locomotion (SAMPL) apparatus, consisting of a chamber where larvae could swim freely, an infrared illuminator, a camera, and software to process video in real time. A comprehensive description of the apparatus is contained in *Zhu et al., 2023*. Here we briefly describe the specific details of our experiments. Larvae were transferred to chambers at densities of 3–8 fish per chamber for 7 dpf experiments or 1–4 fish per chamber for 14 dpf experiments containing 25–30 ml of E3 or 0.2% DMSO / 1 µM capsaicin

for activation experiments. After 24 hr, behavior recordings were paused for 30–60 min for feeding (feeding pause) and 1–2 ml of rotifer culture was added to each chamber. Larvae were removed from the apparatus after 48 hr.

To monitor behavior before/during Purkinje cell activation, 7 dpf larvae were placed in chambers with E3. At 8 and 9 dpf, control fish were placed in 0.2% DMSO in E3 and the condition fish were placed in 1 µM capsaicin in 0.2% DMSO in E3 for 6 hr. The recording started about 10–15 min after adding the fish to the capsaicin solution. Fish were fed after the 6 hr activation period. Video was sampled at 40 Hz in constant darkness. Control: 9626 bouts (63% climb bouts)/149 fish/8 experimental repeats; Activation: 9664 bouts (61% climb bouts)/155 fish/8 experimental repeats.

To monitor behavior before/after Purkinje cell lesions, 7 dpf/14 dpf larvae were placed in the chambers with E3. After feeding at 8 dpf/15 dpf, fish were placed in petri dishes with 0.2% DMSO in E3 (control) or 10 µM capsaicin in 0.2% DMSO in E3 for 40–60 min. Fish were then returned to the chambers in E3 and behavior recording was started. Video was sampled at 160 Hz in constant darkness. 7 dpf lesions: Control: 17895 bouts (61% climb bouts)/110 fish/14 experimental repeats; Lesion: 17819 bouts (57% climb bouts)/120 fish/14 experimental repeats; 14 dpf lesion: Control: 10666 bouts (58% climb bouts)/48 fish/7 experimental repeats; Lesion: 10708 bouts (54% climb bouts)/44 fish/7 experimental repeats.

Pectoral fin amputations were performed at 13 dpf. Two length-matched siblings were anesthetized in 0.2 mg/ml ethyl- 3-aminobenzoic acid ethyl ester (MESAB, catalog # E10521, Sigma-Aldrich) simultaneously and mounted in 2% low-melting temperature agar. Visualized under a stereomicroscope (Leica M80, 20 x/12 eyepieces, 1.0 x objective), the two pectoral fins from one larva were removed by pulling the base of the fin at the scapulocoracoid laterally with #5 Dumont forceps. After amputation, both fish were freed from the agar and allowed to recover in E3 until the next day, at which point half of the amputated and control fish were randomly selected for Purkinje cell lesions. Lesions were performed as above and behavior recorded for 48 hr. Behavior was recorded at a sampling rate of 160 Hz with a 14/10 hr light-dark cycle. Control: 1506/5090/5353 (slow/medium/fast) bouts/15 fish/8 experimental repeats; Purkinje cell lesion: 1667/6166/4299 (slow/medium/fast) bouts/18 fish/8 experimental repeats; Fin amputation: 1935/6295/4911 (slow/medium/fast) bouts/17 fish/8 experimental repeats.

## Behavior analysis

Comprehensive descriptions of behavioral kinematics and baseline data for different genetic backgrounds are detailed in *Zhu et al., 2023*. Here, we describe the specific parameters used for our experiments. Behavior data were analyzed using custom-written software in MATLAB (Mathworks, Natick MA), which extracted individual swim bouts from the raw data (x/z position and pitch angle as a function of time).Only bouts during the circadian day were analyzed. Experimental repeats consisted of data collected across multiple SAMPL boxes from a single clutch of fish; the number of fish available determined how many apparatus were used (1-3). For comparisons across conditions (e.g. activation/control), fish from one clutch were randomly split into control and condition groups. As bout number is the fundamental unit of kinematic analysis, and different numbers of fish available would yield different numbers of bouts, we bounded our experiments to allow comparison across repeats. Specifically, if an experimental repeat contained less than 650 bouts it was excluded.

Between 22% and 27% of lesion experimental repeats contained less than 650 bouts and were not included in the analysis. For the activation experiments 56% (10 of 18) of experimental repeats were excluded with the 650 bouts threshold due to shorter recording times a higher fraction of experiments contained less than the threshold number of bouts. In subsequent analyses, the number of analyzed bouts was matched from both groups for a given experimental repeat to ensure an identical representation of control and condition bouts. Individual bouts were aligned at the time of peak speed. Bouts were excluded if their peak speed was <5 mm/s or the fish rotated more than 30°(120°/s) during the acceleration. The fractions excluded were as follows: for 7 dpf ablation: ctrl 0.2% lesion 0.15%; 7 dpf activation: ctrl 1% activation 1.7%; 14 dpf ablation dark: ctrl 0.05% ablation 0.05%; 14 dpf ablation light: ctrl 0.02% ablation 0.02%. For each experiment between 0.02% and 1.7% of bouts were excluded based on those criteria. Data was recorded either at 40 Hz (activation experiments) or 160 Hz (all other experiments). Effect size was calculated as the difference between the control value and the condition value relative to the control value. For fin body slope effect size the control value of

the fast bin (i.e. largest slope) was used for effect size calculations to avoid overestimation of changes due to small control values.

Kinematic analyses proceeded as in *Zhu et al., 2023*; key parameters were defined as follows:

- Posture is the pitch angle of the fish (long axis of the body relative to the horizon) at –250ms relative to peak speed, just before swim bout initiation. Positive values are nose-up.
- Climb Bouts are bouts with a trajectory of > 0°at the peak speed of the swim bout.
- Dive Bouts are bouts with a trajectory of < 0°at the peak speed of the swim bout.
- Upward rotation refers to the rotation from –250ms to the peak angular velocity; only bouts with positive upward rotation were included in the analysis of fin-body coordination.
- Lift is the residual change in depth (z) across a bout after subtracting the change expected from the posture of the fish as detailed in *Ehrlich and Schoppik, 2019*. Briefly, the expected change is calculated using the distance the fish moves in x from –100 to 100ms and the pitch angle at –100ms. Only bouts with positive lift were included in the analysis of fin-body coordination.
- Fin-lift/rotation coordination is defined as the slope of the best linear fit between upward rotation and lift across bouts. The goodness of fit, R2 was used as a measure of how well the fins and trunk are coordinated to generate lift, after *Ehrlich and Schoppik, 2019*.

## Functional GCaMP imaging in Purkinje cells

All calcium imaging experiments were performed using Tilt In Place Microscopy (TIPM), described comprehensively in *Hamling et al., 2023b*. Briefly, 7 dpf fish were mounted in the center of the uncoated side of a mirror galvanometer (catalog #GVS0111, Thorlabs) in 2% low-melting- point agarose. E3 was placed over the agarose, and the galvanometer mirror was placed under the microscope.

A microscope (Thorlabs Bergamo) was used to measure fluorescence elicited by multiphoton excitation (920 nm) from a pulsed infrared laser (Mai Tai HP). Fast volumetric scanning was achieved using a piezo actuator (catalog #PFM450E, Thorlabs) to move the objective.Each frame of the volume (224x96 pixels) was collected with a 0.6 μs pixel dwell time (19.1 frames/s) resulting in a sampling rate of 3.82 volumes/s. While this imaging rate might be too slow to distinguish single spikes, it is suitable to measure a difference in calcium transients upon pitch stimulation to nose-up or nose-down direction. To set the galvanometer to a specific angle, a corresponding voltage was applied. The total angular range of the galvanometer is 40°. For the 30°stimuli, the galvanometer was driven to either +15° or –15° and then rotated so that the mirror was horizontal which allowed allowing for a 30° deflection in one direction. For each cell, 21 trials were initially recorded for one stimulus direction. The galvanometer was then remounted to allow for a 30° stimulus in the opposite direction, and 21 trials were similarly recorded for this direction. The order of nose-up and nose-down blocks were alternated for different fish. After all 42 trials were recorded fish were anesthetized with 0.2 mg/ml MESAB; after 10 min the baseline fluorescence at ±30° was recorded to establish a baseline that controlled for eccentricity. Analysis was done using Fiji and MATLAB. In total, 43 Purkinje cells were imaged and 31 cells were kept from 8 fish. Only Purkinje cells that could be reliably identified at ±30° were analyzed. To map the anatomical locations of the recorded cells, we imaged overview stacks for each fish. These stacks were manually aligned in Illustrator, and the cells included in the analysis were identified and color-coded according to their tuning properties.

Regions of interest were drawn in Fiji and loaded into MATLAB to extract the intensity of fluorescence after motion correction was performed (*Pnevmatikakis and Giovannucci, 2017*). The integral of each stimulus was calculated and trials of the same direction were averaged as the tonic response to ±30°pitch. To extract cells with directional information the directionality index (DI) was calculated by dividing the difference of the up and down responses by the sum of it. Cells with a DI greater than ± 0.35 were considered directionally tuned. Only Purkinje cells that were not directionally tuned were used for PCA and subsequent support vector machine decoding analysis.

To classify trial identity in the dataset, we used a support vector machine (SVM) with a linear kernel. The SVM model was trained using k-fold cross-validation, which splits the data into k subsets (folds). At each iteration, the model was trained on k-1 folds and tested on the remaining fold, ensuring that the model performance was evaluated on unseen data in each fold. Permutations were performed on randomized trial identity as a null hypothesis (fivefold cross-validation; 100 shuffles for randomization). Accuracy was calculated as 1 minus the classification loss.

For calcium imaging in 7 and 14 dpf larvae, a horizontal imaging protocol was used. In total 11 fish were imaged at 7 dpf and 7 fish at 14 dpf. A total of 138/90 (7/14 dpf) cells were recorded. Cells were imaged while the fish was horizontal. For horizontal imaging, we used a ±19° stimulus, enabling us to alternate between up and down trials without the need to remount the galvanometer. Before each trial, a 15 s period was recorded; the average activity during this time was used as the baseline. Fish were pitched nose-down (–19°) for 15 s and rapidly returned to horizontal, whereupon calcium activity was measured. This stimulus was then repeated in the nose-up (+19°) direction. The maximum dFF of the first second upon return was analyzed. Cells were classified into directional or non-directional based on the directionality index as described above. PCA and decoder analyses were performed using activity from non-directional cells. Decoding accuracy was tested for each fish individually.

## Statistics

All statistical testing was done in Matlab R2020a. Data across repeats was pooled for analysis. To assess the variability and determine whether pooling individual experimental repeats within each group was appropriate, we performed a two-way analysis of variance (ANOVA) on the interquartile ranges (IQRs) of the single experimental repeats for the 7 days post-fertilization (dpf) activation, the 7dpf lesion, and the 14dpf lesion experiments without excluding experimental repeats. The results of the ANOVAs and the IQRs for all experimental repeats are reported in *Appendix 1—tables 6–11*.

To estimate the spread of the data we resampled distributions 100 times with replacement from the data from each condition and computed the expected value for control and perturbed datasets. These permutations were then used to explicitly compute a p-value for fitted variables (slope and $R^2$ of fin body coordination).

For other variables two-sided Wilcoxon rank sum tests were performed. To correct for multiple testing the critical p-value was calculated based on $\alpha = 0.05$ using Šidák's method. The critical p-value for each data set is reported in the respective table. Outliers were determined as deviating more than three times the scaled median absolute deviation (MAD) from the median. A scaling factor of 1.4826 was used to ensure that MAD-based outlier detection is consistent with other methods like Z-scores. Data is shown as median and 95% confidence interval of the median for measured parameters or as median with 25th and 75th percentile for bootstrapped variables. The 95% confidence intervals of the median were bootstrapped using 1000 samples. The medians with 95% confidence intervals for all parameters are reported in the tables. For linear fits a robust regression model (bisquare) was used and fitted variables (slope and $R^2$ of fin body coordination) were bootstrapped (using 100 samples). To test speed dependency of the fin-lift/rotation ratio Spearman's rank correlation was computed and control and lesion values were compared. First, the correlation coefficients where transformed using Fisher's z-transformation to enable direct comparison of the z-scores. The difference between the z-scores was divided by the standard error and a z-test was performed. Additionally, we only considered effect sizes of ≥15% to be biologically relevant.

## Acknowledgements

*Tg(aldoca:TRPV1-tagRFP)* fish were generated using a plasmid that was a gift from David Prober's laboratory. Research was supported by the National Institute on Deafness and Communication Disorders of the National Institutes of Health under award number R01DC017489. The authors would like to thank Martha Bagnall and Abigail Person along with the members of the Schoppik and Nagel lab for their valuable feedback and discussions.

## Additional information

### Funding

| Funder | Grant reference number | Author |
| --- | --- | --- |
| National Institute on Deafness and Other Communication Disorders | R01DC017489 | David Schoppik |

| Funder | Grant reference number | Author |
|--------|------------------------|--------|

The funders had no role in study design, data collection and interpretation, or the decision to submit the work for publication.

## Author contributions

Franziska Auer, Conceptualization, Funding acquisition, Investigation, Visualization, Writing – original draft; Katherine Nardone, Investigation; Koji Matsuda, Tg(aldoca:TRPV1-tagRFP) fish generation; Masahiko Hibi, Supervision; David Schoppik, Conceptualization, Supervision, Funding acquisition, Methodology, Writing – original draft, Writing – review and editing

## Author ORCIDs

Franziska Auer ![ORCID] https://orcid.org/0000-0002-4389-9963
Masahiko Hibi ![ORCID] https://orcid.org/0000-0002-9142-4444
David Schoppik ![ORCID] https://orcid.org/0000-0001-7969-9632

## Ethics

All procedures involving zebrafish larvae were approved by the Institutional Animal Care and Use Committee of New York University.

Reviewer #1 (Public review): https://doi.org/10.7554/eLife.97614.3.sa1
Reviewer #2 (Public review): https://doi.org/10.7554/eLife.97614.3.sa2
Reviewer #3 (Public review): https://doi.org/10.7554/eLife.97614.3.sa3
Author response https://doi.org/10.7554/eLife.97614.3.sa4

# Additional files

## Supplementary files

MDAR checklist

## Data availability

All data, raw and analyzed, as well as code necessary to generate the figures is available at the following https://doi.org/10.17605/OSF.IO/9X57Z.

The following dataset was generated:

| Author(s) | Year | Dataset title | Dataset URL | Database and Identifier |
|-----------|------|---------------|-------------|-------------------------|
| Auer F, Nardone K, Matsuda K, Hibi M, Schoppik D | 2025 | Data associated with "Purkinje cells control balance in larval zebrafish" | https://doi.org/10.17605/OSF.IO/9X57Z | Open Science Framework, 10.17605/OSF.IO/9X57Z |

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

# Appendix 1

**Appendix 1—table 1.** Behavior measurements 7 dpf Purkinje cell activation.

| | control median [95% CI] | activation median [95% CI] | effect [%] | p-value | significance |
|---|---|---|---|---|---|
| pre activation | | | | | |
| critical p-value: 0.006 | | | | | |
| climb posture [°] | 15.4 [15.1–15.8] | 16.1 [15.6–16.4] | 4 | 0.006 | no |
| dive posture [°] | −12.9 [-13.2–-12.6] | −13.5 [-13.8–-13.3] | 5 | <0.001 | no |
| bout duration [s] | 0.2 [0.2–0.2] | 0.2 [0.2–0.2] | 0 | <0.001 | no |
| Inter-bout interval [s] | 1.4 [1.4–1.4] | 1.4 [1.4–1.4] | 2 | <0.001 | no |
| speed [mm/s] | 12.5 [12.4–12.6] | 12.6 [12.5–12.6] | 0 | 0.831 | no |
| slope slow [mm/°] | 0.014 [0.014–0.014] | 0.012 [0.012–0.012] | -4 | 0.078 | no |
| slope medium [mm/°] | 0.017 [0.016–0.017] | 0.016 [0.016–0.016] | -1 | 0.352 | no |
| slope fast [mm/°] | 0.041 [0.041–0.041] | 0.042 [0.041–0.042] | 1 | 0.385 | no |
| post activation | | | | | |
| critical p-value: 0.006 | | | | | |
| climb posture [°] | 14.7 [14.0–15.4] | 19.0 [18.5–19.7] | 29 | <0.001 | yes |
| dive posture [°] | −16.6 [-16.9–-16.1] | −20.5 [-20.9–-20.1] | 24 | <0.001 | yes |
| bout duration [s] | 0.2 [0.2–0.2] | 0.2 [0.2–0.2] | 0 | <0.001 | no |
| Inter-bout interval [s] | 1.6 [1.5–1.6] | 1.6 [1.5–1.6] | 2 | 0.043 | no |
| speed [mm/s] | 12.7 [12.5–12.8] | 13.0 [12.9–13.1] | 3 | <0.001 | no |
| slope slow [mm/°] | 0.012 [0.011–0.012] | 0.009 [0.009–0.010] | -6 | 0.149 | no |
| slope medium [mm/°] | 0.019 [0.018–0.019] | 0.015 [0.015–0.015] | −11 | <0.001 | no |
| slope fast [mm/°] | 0.037 [0.037–0.038] | 0.034 [0.034–0.035] | -8 | 0.069 | no |

**Appendix 1—table 2.** Behavior measurements 7 dpf Purkinje cell lesion.

| | control median [95% CI] | lesion median [95% CI] | effect [%] | p-value | significance |
|---|---|---|---|---|---|
| pre lesion | | | | | |
| critical p-value: 0.006 | | | | | |
| climb posture [°] | 18.0 [17.6–18.4] | 19.0 [18.6–19.4] | 6 | 0.001 | no |
| dive posture [°] | −11.9 [-12.1 to -11.6] | −11.5 [-11.7–-11.3] | -3 | 0.25 | no |
| bout duration [s] | 0.2 [0.2–0.2] | 0.2 [0.2–0.2] | 4 | 0.001 | no |
| Inter-bout interval [s] | 1.8 [1.8–1.9] | 1.7 [1.7–1.8] | -4 | <0.001 | no |
| speed [mm/s] | 11.3 [11.2, 11.4] | 12.0 [11.9, 12.1] | 6 | <0.001 | no |
| slope slow [mm/°] | 0.003 [0.003–0.003] | 0.004 [0.004–0.004] | 2 | 0.261 | no |
| slope medium [mm/°] | 0.008 [0.008–0.008] | 0.001 [0.001–0.002] | −14 | <0.001 | no |
| slope fast [mm/°] | 0.050 [0.049–0.050] | 0.052 [0.051–0.053] | 5 | 0.284 | no |
| post lesion | | | | | |
| critical p-value: 0.006 | | | | | |
| climb posture [°] | 10.0 [9.5–10.7] | 13.6 [13.1–14.3] | 36 | <0.001 | yes |
| dive posture [°] | −11.7 [-11.9–-11.5] | −11.2 [-11.4–-11.0] | -4 | 0.002 | no |

*Appendix 1—table 2 Continued on next page*

*Appendix 1—table 2 Continued*

|  | control median [95% CI] | lesion median [95% CI] | effect [%] | p-value | significance |
|---|---|---|---|---|---|
| bout duration [s] | 0.2 [0.2–0.2] | 0.1 [0.1–0.1] | -4 | <0.001 | no |
| Inter-bout interval [s] | 1.7 [1.7–1.8] | 1.7 [1.7–1.7] | -2 | 0.203 | no |
| speed [mm/s] | 10.3 [10.2–10.4] | 10.6 [10.5–10.7] | 2 | <0.001 | no |
| slope slow [mm/°] | 0.008 [0.007–0.008] | 0.005 [0.005–0.005] | -6 | 0.002 | no |
| slope medium [mm/°] | 0.012 [0.012–0.012] | 0.008 [0.007–0.008] | −10 | <0.001 | no |
| slope fast [mm/°] | 0.047 [0.047–0.048] | 0.025 [0.025–0.026] | −46 | <0.001 | yes |

**Appendix 1—table 3.** Behavior measurements 14 dpf Purkinje cell lesion.

|  | control median [95% CI] | lesion median [95% CI] | effect [%] | p-value | significance |
|---|---|---|---|---|---|
| pre lesion |  |  |  |  |  |
| critical p-value: 0.01 |  |  |  |  |  |
| climb posture [°] | 16.0 [15.6–16.5] | 15.2 [14.7–15.9] | -5 | <0.001 | no |
| dive posture [°] | −9.0 [-9.4–-8.7] | −9.5 [-9.7–-9.1] | 5 | 0.006 | no |
| bout duration [s] | 0.2 [0.2–0.2] | 0.2 [0.2–0.2] | 3 | 0.59 | no |
| Inter-bout interval [s] | 2.3 [2.3–2.4] | 2.4 [2.4–2.4] | 4 | 0.29 | no |
| speed [mm/s] | 10.2 [10.0–10.3] | 10.5 [10.4–10.7] | 4 | <0.001 | no |
| post lesion |  |  |  |  |  |
| critical p-value: 0.01 |  |  |  |  |  |
| climb posture [°] | 14.3 [13.8–14.8] | 17.1 [16.2–17.8] | 20 | <0.001 | yes |
| dive posture [°] | −9.8 [-10.1–-9.5] | −12.3 [-12.6–-11.9] | 26 | <0.001 | yes |
| bout duration [s] | 0.2 [0.2–0.2] | 0.2 [0.2–0.2] | -8 | <0.001 | no |
| Inter-bout interval [s] | 2.9 [2.8–3.0] | 2.8 [2.7–2.8] | -4 | 0.01 | no |
| speed [mm/s] | 9.7 [9.6–9.8] | 9.3 [9.2–9.4] | -3 | <0.001 | no |

**Appendix 1—table 4.** Behavior measurements 14 dpf Purkinje cell lesion.

|  | control median [95% CI] | lesion median [95% CI] | effect [%] | p-value | significance |
|---|---|---|---|---|---|
| lesion |  |  |  |  |  |
| critical p-value: 0.004 |  |  |  |  |  |
| slope slow [mm/°] | 0.029 [0.028–0.031] | 0.033 [0.032–0.034] | 6 | 0.341 | no |
| slope medium [mm/°] | 0.041 [0.040–0.044] | 0.018 [0.018–0.019] | −34 | <0.001 | yes |
| slope fast [mm/°] | 0.068 [0.067–0.071] | 0.026 [0.026–0.027] | −62 | <0.001 | yes |
| $R^2$slow | 0.212 [0.201–0.227] | 0.371 [0.361–0.392] | 35 | 0.005 | no |
| $R^2$ medium | 0.382 [0.373–0.388] | 0.603 [0.594–0.611] | 48 | <0.001 | yes |
| $R^2$ fast | 0.460 [0.452–0.471] | 0.641 [0.634–0.648] | 39 | 0.001 | yes |
| Fin lift slow [mm] | 0.184 [0.168–0.199] | 0.164 [0.155–0.172] | −11 | 0.004 | no |
| Fin lift medium [mm] | 0.201 [0.192–0.208] | 0.178 [0.171–0.185] | −11 | 0.002 | no |
| Fin lift fast [mm] | 0.267 [0.258–0.277] | 0.189 [0.174–0.197] | −29 | <0.001 | yes |
| Rotation slow [°] | 1.471 [1.348–1.686] | 1.254 [1.140–1.383] | −15 | 0.005 | no |
| Rotation medium [°] | 1.265 [1.176–1.324] | 1.178 [1.120–1.275] | -7 | 0.515 | no |
| Rotation fast [°] | 1.083 [1.018–1.150] | 1.241 [1.180–1.336] | 15 | <0.001 | no |

**Appendix 1—table 5.** Behavior measurements 14 dpf pectoral fin amputation.

| | control median [95% CI] | fin amputation median [95% CI] | effect [%] | p-value | significance |
|---|---|---|---|---|---|
| fin amputation | | | | | |
| critical p-value: 0.004 | | | | | |
| slope slow [mm/°] | 0.036 [0.035–0.037] | –0.005 [-0.005–-0.004] | –60 | <0.001 | yes |
| slope medium [mm/°] | 0.055 [0.054–0.056] | –0.005 [-0.005–-0.005] | –88 | <0.001 | yes |
| slope fast [mm/°] | 0.068 [0.067–0.069] | 0.013 [0.013–0.013] | –81 | <0.001 | yes |
| $R^2$ slow | 0.173 [0.162–0.187] | 0.017 [0.003–0.037] | –14 | 0.141 | no |
| $R^2$ medium | 0.370 [0.365–0.379] | 0.011 [0.003–0.019] | –61 | <0.001 | yes |
| $R^2$ fast | 0.450 [0.441–0.458] | 0.090 [0.053–0.144] | –26 | <0.0388 | no |
| Fin lift slow [mm] | 0.206 [0.196–0.222] | 0.066 [0.058–0.075] | –68 | <0.001 | yes |
| Fin lift medium [mm] | 0.219 [0.209–0.226] | 0.082 [0.078–0.087] | –62 | <0.001 | yes |
| Fin lift fast [mm] | 0.275 [0.263–0.284] | 0.123 [0.117–0.129] | –55 | <0.001 | yes |
| Rotation slow [°] | 1.544 [1.402–1.689] | 0.546 [0.455–0.652] | –65 | <0.001 | yes |
| Rotation medium [°] | 1.339 [1.278–1.419] | 0.642 [0.609–0.689] | –52 | <0.001 | yes |
| Rotation fast [°] | 1.046 [0.975–1.119] | 0.950 [0.887–1.033] | -9 | 0.014 | no |

**Appendix 1—table 6.** Results of ANOVA on interquartile ranges (IQRs) of single experimental repeats for 7 Days Post-Fertilization (dpf) activation experiments.

| Source | Sum Sq. | d.f. | Mean Sq. | F | Prob >F |
|---|---|---|---|---|---|
| Group | 65.0 | 3.0 | 21.7 | 0.225 | 0.879 |
| Measurement Type | 34.6 | 4.0 | 8.7 | 0.090 | 0.986 |
| Group*Measurement Type | 80.3 | 12.0 | 6.7 | 0.069 | 1.000 |
| Error | 32818.0 | 340.0 | 96.5 | | |
| Total | 32998.0 | 359.0 | | | |

**Appendix 1—table 7.** Results of ANOVA on interquartile ranges (IQRs) of single experimental repeats for 7 Days Post-Fertilization (dpf) lesion experiments.

| Source | Sum Sq. | d.f. | Mean Sq. | F | Prob >F |
|---|---|---|---|---|---|
| Group | 15.0 | 3 | 5.0 | 0.064 | 0.979 |
| Measurement Type | 70.0 | 4 | 17.5 | 0.223 | 0.925 |
| Group*Measurement Type | 34.2 | 12 | 2.9 | 0.036 | 1.000 |
| Error | 25069.4 | 320 | 78.3 | | |
| Total | 25188.6 | 339 | | | |

**Appendix 1—table 8.** Results of ANOVA on interquartile ranges (IQRs) of single experimental repeats for 14 Days Post-Fertilization (dpf) lesion experiments.

| Source | Sum Sq. | d.f. | Mean Sq. | F | Prob >F |
|---|---|---|---|---|---|
| Group | 8.3 | 3.0 | 2.8 | 0.028 | 0.994 |
| Measurement Type | 382.8 | 4.0 | 95.7 | 0.973 | 0.423 |

*Appendix 1—table 8 Continued on next page*

*Appendix 1—table 8 Continued*

| Source | Sum Sq. | d.f. | Mean Sq. | F | Prob >F |
|---|---|---|---|---|---|
| Group*Measurement Type | 47.3 | 12.0 | 3.9 | 0.040 | 1.000 |
| Error | 17700.5 | 180.0 | 98.3 | | |
| Total | 18138.9 | 199.0 | | | |

**Appendix 1—table 9.** IQR for all experimental repeats prior to excluding any repeats for 7dpf Purkinje cell activation data set.

| | climb bout posture [°] | dive bout posture [°] | Duration [s] | IBI [s] | Speed [mm/s] |
|---|---|---|---|---|---|
| | 20.74 | 23.65 | 0.075 | 1.33 | 7.36 |
| | 28.21 | 16.96 | 0.125 | 1.00 | 7.28 |
| | 29.90 | 24.49 | 0.125 | 1.40 | 9.43 |
| | 24.84 | 17.06 | 0.075 | 1.55 | 7.96 |
| | 22.60 | 18.18 | 0.075 | 1.60 | 7.64 |
| | 23.63 | 15.56 | 0.075 | 2.04 | 4.99 |
| | 18.86 | 17.54 | 0.100 | 2.48 | 6.69 |
| | 22.52 | 15.44 | 0.075 | 2.03 | 6.93 |
| | 23.00 | 11.93 | 0.081 | 2.38 | 6.92 |
| | 20.05 | 14.85 | 0.075 | 1.61 | 7.30 |
| | 19.29 | 18.75 | 0.100 | 2.30 | 6.00 |
| | 22.25 | 14.81 | 0.075 | 2.23 | 6.25 |
| | 21.61 | 17.06 | 0.075 | 1.95 | 6.36 |
| | 24.16 | 18.94 | 0.100 | 1.68 | 6.66 |
| | 19.63 | 10.03 | 0.050 | 1.75 | 6.68 |
| | 22.35 | 16.96 | 0.100 | 1.90 | 6.44 |
| | 21.22 | 13.89 | 0.100 | 2.50 | 7.08 |
| pre control | 21.61 | 14.43 | 0.075 | 1.68 | 7.10 |

*Appendix 1—table 9 Continued on next page*

*Appendix 1—table 9 Continued*

| | climb bout posture [°] | dive bout posture [°] | Duration [s] | IBI [s] | Speed [mm/s] |
|---|---|---|---|---|---|
| | 24.68 | 16.86 | 0.081 | 1.87 | 7.44 |
| | 25.48 | 13.85 | 0.100 | 1.30 | 7.71 |
| | 25.88 | 13.53 | 0.100 | 1.08 | 6.48 |
| | 24.92 | 16.52 | 0.075 | 1.68 | 7.13 |
| | 20.30 | 18.64 | 0.075 | 1.38 | 4.41 |
| | 23.32 | 21.46 | 0.100 | 2.99 | 6.39 |
| | 25.03 | 15.73 | 0.100 | 1.83 | 6.68 |
| | 21.49 | 14.53 | 0.100 | 2.12 | 6.38 |
| | 25.84 | 16.73 | 0.100 | 1.58 | 6.66 |
| | 22.33 | 12.52 | 0.075 | 2.10 | 6.67 |
| | 20.44 | 16.83 | 0.075 | 3.10 | 6.05 |
| | 23.67 | 16.49 | 0.100 | 2.78 | 6.38 |
| | 27.07 | 19.06 | 0.100 | 2.70 | 6.32 |
| | 23.81 | 17.40 | 0.075 | 1.66 | 7.31 |
| | 21.82 | 19.38 | 0.050 | 2.33 | 7.68 |
| | 24.27 | 16.03 | 0.100 | 2.33 | 6.73 |
| | 23.44 | 16.06 | 0.100 | 1.98 | 7.12 |
| pre activation | 22.89 | 15.45 | 0.100 | 1.58 | 6.93 |
| | 21.10 | 16.75 | 0.050 | 0.43 | 5.69 |
| | 24.43 | 43.20 | 0.100 | 1.40 | 7.61 |
| | 36.40 | 18.92 | 0.100 | 1.56 | 8.08 |
| | 21.95 | 18.46 | 0.075 | 1.96 | 7.08 |
| | 21.82 | 20.37 | 0.075 | 1.25 | 8.00 |
| | 20.74 | 19.04 | 0.075 | 1.98 | 6.57 |
| | 22.37 | 20.55 | 0.075 | 1.53 | 6.95 |
| | 21.79 | 13.07 | 0.075 | 2.05 | 7.40 |
| | 23.97 | 16.79 | 0.075 | 2.43 | 6.41 |
| | 22.48 | 16.57 | 0.075 | 2.09 | 6.38 |
| | 21.13 | 21.03 | 0.075 | 2.33 | 6.43 |
| | 23.19 | 17.84 | 0.075 | 1.98 | 6.47 |
| | 22.40 | 18.91 | 0.075 | 1.73 | 6.46 |
| | 25.41 | 20.43 | 0.075 | 2.08 | 6.83 |
| | 29.93 | 18.46 | 0.075 | 1.90 | 5.79 |
| | 24.37 | 25.53 | 0.100 | 1.80 | 6.58 |
| | 26.21 | 19.68 | 0.100 | 1.53 | 6.10 |
| control | 20.50 | 18.00 | 0.075 | 1.50 | 6.32 |

*Appendix 1—table 9 Continued on next page*

*Appendix 1—table 9 Continued*

| | climb bout posture [°] | dive bout posture [°] | Duration [s] | IBI [s] | Speed [mm/s] |
|---|---|---|---|---|---|
| | 23.40 | 28.68 | 0.075 | 0.65 | 6.53 |
| | 26.09 | 11.16 | 0.075 | 1.65 | 7.51 |
| | 21.35 | 19.42 | 0.050 | 0.75 | 6.68 |
| | 23.69 | 13.84 | 0.075 | 2.31 | 6.34 |
| | 19.90 | 15.18 | 0.050 | 1.25 | 5.33 |
| | 24.64 | 20.85 | 0.050 | 1.96 | 6.96 |
| | 23.64 | 22.08 | 0.075 | 1.78 | 7.22 |
| | 21.78 | 18.14 | 0.100 | 2.19 | 6.98 |
| | 24.24 | 17.35 | 0.100 | 2.57 | 8.79 |
| | 27.93 | 13.88 | 0.100 | 2.63 | 7.56 |
| | 23.48 | 21.20 | 0.075 | 2.51 | 6.25 |
| | 26.16 | 20.58 | 0.100 | 2.33 | 6.95 |
| | 27.49 | 26.34 | 0.081 | 1.73 | 6.60 |
| | 26.88 | 23.40 | 0.075 | 1.85 | 7.78 |
| | 25.26 | 19.62 | 0.075 | 2.10 | 7.60 |
| | 24.41 | 20.94 | 0.100 | 2.33 | 6.47 |
| | 26.64 | 19.76 | 0.100 | 1.90 | 6.69 |
| activation | 23.86 | 19.33 | 0.075 | 1.53 | 6.45 |

**Appendix 1—table 10.** IQR for all experimental repeats prior to excluding any repeats for 7dpf Purkinje cell lesion data set.

| | climb bout posture [°] | dive bout posture [°] | Duration [s] | IBI [s] | Speed [mm/s] |
|---|---|---|---|---|---|
| | 24.53 | 14.18 | 0.113 | 3.67 | 9.85 |
| | 29.07 | 16.59 | 0.088 | 5.22 | 6.61 |
| | 23.45 | 16.59 | 0.094 | 6.22 | 7.99 |
| | 25.97 | 15.89 | 0.125 | 3.73 | 9.84 |
| | 20.95 | 15.60 | 0.094 | 3.48 | 6.82 |
| | 19.82 | 19.78 | 0.100 | 3.29 | 9.77 |
| | 23.70 | 13.59 | 0.119 | 5.99 | 7.03 |
| | 24.08 | 18.19 | 0.095 | 3.87 | 8.89 |
| | 26.89 | 12.68 | 0.088 | 2.70 | 6.42 |
| | 22.21 | 14.27 | 0.113 | 5.84 | 7.13 |
| | 21.91 | 17.16 | 0.113 | 2.49 | 9.02 |
| | 24.20 | 14.05 | 0.131 | 7.95 | 6.71 |
| | 20.42 | 16.63 | 0.088 | 2.80 | 5.97 |
| | 18.70 | 11.07 | 0.094 | 3.17 | 6.10 |
| | 26.29 | 10.92 | 0.094 | 2.18 | 6.52 |
| | 23.38 | 11.81 | 0.063 | 1.03 | 7.63 |
| pre control | 18.46 | 13.09 | 0.106 | 1.73 | 7.54 |

*Appendix 1—table 10 Continued on next page*

*Appendix 1—table 10 Continued*

|  | climb bout posture [°] | dive bout posture [°] | Duration [s] | IBI [s] | Speed [mm/s] |
|---|---|---|---|---|---|
|  | 16.27 | 19.05 | 0.098 | 1.64 | 7.82 |
|  | 22.59 | 14.01 | 0.113 | 6.83 | 7.65 |
|  | 27.26 | 17.13 | 0.088 | 3.37 | 8.68 |
|  | 26.93 | 20.38 | 0.119 | 2.15 | 8.99 |
|  | 26.21 | 12.07 | 0.088 | 4.38 | 4.64 |
|  | 28.53 | 14.25 | 0.094 | 3.81 | 7.55 |
|  | 22.40 | 14.02 | 0.106 | 5.10 | 7.70 |
|  | 21.12 | 15.35 | 0.106 | 3.70 | 9.32 |
|  | 24.34 | 13.21 | 0.100 | 2.75 | 7.15 |
|  | 21.10 | 15.76 | 0.094 | 4.42 | 7.88 |
|  | 25.81 | 15.31 | 0.100 | 4.41 | 8.61 |
|  | 30.09 | 13.90 | 0.119 | 5.43 | 7.73 |
|  | 21.85 | 12.45 | 0.106 | 4.18 | 7.82 |
|  | 19.56 | 18.35 | 0.072 | 2.10 | 5.06 |
|  | 20.57 | 10.43 | 0.091 | 2.39 | 6.44 |
|  | 21.79 | 15.38 | 0.075 | 2.69 | 6.59 |
| pre lesion | 22.69 | 16.77 | 0.109 | 2.09 | 9.98 |
|  | 22.43 | 24.44 | 0.100 | 2.29 | 9.60 |
|  | 21.46 | 16.25 | 0.106 | 6.89 | 4.19 |
|  | 29.27 | 14.18 | 0.094 | 3.72 | 7.25 |
|  | 22.57 | 14.49 | 0.069 | 3.55 | 7.13 |
|  | 20.13 | 16.04 | 0.100 | 4.93 | 6.48 |
|  | 23.73 | 17.81 | 0.088 | 3.65 | 6.99 |
|  | 25.54 | 12.29 | 0.119 | 4.97 | 5.39 |
|  | 16.56 | 14.54 | 0.088 | 3.40 | 10.50 |
|  | 25.93 | 13.75 | 0.094 | 4.08 | 5.71 |
|  | 24.72 | 15.81 | 0.119 | 6.18 | 5.81 |
|  | 23.33 | 10.90 | 0.094 | 2.99 | 4.77 |
|  | 23.03 | 11.87 | 0.106 | 7.14 | 5.50 |
|  | 20.11 | 13.99 | 0.088 | 3.59 | 6.06 |
|  | 18.18 | 18.27 | 0.091 | 2.46 | 5.63 |
|  | 21.23 | 13.98 | 0.063 | 0.85 | 4.65 |
|  | 20.15 | 13.44 | 0.094 | 1.09 | 5.63 |
| post control | 20.30 | 13.38 | 0.063 | 0.84 | 5.27 |

*Appendix 1—table 10 Continued on next page*

*Appendix 1—table 10 Continued*

| | climb bout posture [°] | dive bout posture [°] | Duration [s] | IBI [s] | Speed [mm/s] |
|---|---|---|---|---|---|
| | 18.49 | 13.28 | 0.063 | 5.34 | 5.32 |
| | 24.95 | 12.55 | 0.113 | 7.19 | 4.72 |
| | 28.40 | 15.31 | 0.078 | 4.29 | 7.82 |
| | 27.02 | 11.85 | 0.106 | 6.30 | 6.02 |
| | 23.87 | 11.08 | 0.063 | 5.39 | 5.24 |
| | 23.90 | 12.96 | 0.069 | 1.96 | 6.24 |
| | 26.79 | 11.18 | 0.069 | 0.91 | 5.87 |
| | 26.27 | 15.42 | 0.094 | 4.43 | 8.02 |
| | 25.66 | 14.43 | 0.100 | 6.43 | 5.88 |
| | 24.62 | 13.05 | 0.125 | 6.24 | 5.66 |
| | 25.76 | 12.31 | 0.100 | 4.98 | 7.33 |
| | 28.85 | 12.47 | 0.094 | 6.11 | 5.98 |
| | 22.35 | 13.41 | 0.075 | 1.85 | 6.94 |
| | 22.36 | 17.24 | 0.088 | 3.17 | 5.22 |
| | 22.52 | 14.74 | 0.069 | 0.88 | 5.07 |
| | 27.76 | 12.45 | 0.094 | 2.01 | 4.49 |
| post lesion | 24.21 | 12.28 | 0.056 | 1.16 | 4.98 |

**Appendix 1—table 11.** IQR for all experimental repeats prior to excluding any repeats for 14dpf Purkinje cell lesion data set.

| | climb bout posture [°] | dive bout posture [°] | Duration [s] | IBI [s] | Speed [mm/s] |
|---|---|---|---|---|---|
| | 28.70 | 14.05 | 0.136 | 5.04 | 7.83 |
| | 26.16 | 14.40 | 0.113 | 4.06 | 7.27 |
| | 35.74 | 18.05 | 0.119 | 2.53 | 4.47 |
| | 30.01 | 23.57 | 0.125 | 1.26 | 3.32 |
| | 30.92 | 18.92 | 0.175 | 3.28 | 6.19 |
| | 29.46 | 15.84 | 0.194 | 3.83 | 7.89 |
| | 6.29 | 9.94 | 0.169 | 1.38 | 4.81 |
| | 15.57 | 7.98 | 0.131 | 3.11 | 5.13 |
| | 20.51 | 13.12 | 0.138 | 4.17 | 7.72 |
| pre control | 20.14 | 15.36 | 0.169 | 3.26 | 6.63 |

*Appendix 1—table 11 Continued on next page*

*Appendix 1—table 11 Continued*

|  | climb bout posture [°] | dive bout posture [°] | Duration [s] | IBI [s] | Speed [mm/s] |
|---|---|---|---|---|---|
|  | 24.18 | 13.16 | 0.131 | 6.20 | 5.15 |
|  | 19.37 | 18.62 | 0.138 | 2.32 | 7.81 |
|  | 33.27 | 18.72 | 0.122 | 3.30 | 5.90 |
|  | 21.64 | 14.69 | 0.128 | 2.60 | 5.18 |
|  | 32.50 | 16.35 | 0.169 | 3.91 | 6.48 |
|  | 28.20 | 17.16 | 0.169 | 1.68 | 6.59 |
|  | 10.11 | 6.42 | 0.150 | 3.91 | 4.40 |
|  | 23.29 | 17.03 | 0.100 | 2.95 | 5.89 |
|  | 30.73 | 10.10 | 0.150 | 3.95 | 8.29 |
| pre lesion | 22.80 | 14.67 | 0.156 | 4.61 | 8.10 |
|  | 26.02 | 13.49 | 0.119 | 4.14 | 7.82 |
|  | 24.71 | 14.18 | 0.100 | 5.50 | 7.34 |
|  | 32.91 | 16.37 | 0.125 | 5.63 | 5.88 |
|  | 26.24 | 21.08 | 0.100 | 1.75 | 3.42 |
|  | 26.15 | 18.88 | 0.119 | 3.98 | 5.59 |
|  | 34.05 | 17.22 | 0.131 | 5.06 | 5.75 |
|  | 4.98 | 9.41 | 0.100 | 1.74 | 4.02 |
|  | 10.19 | 15.13 | 0.088 | 5.12 | 4.07 |
|  | 24.15 | 12.57 | 0.144 | 4.92 | 7.42 |
| post control | 27.81 | 18.68 | 0.144 | 3.74 | 6.66 |
|  | 21.22 | 13.02 | 0.106 | 6.42 | 4.59 |
|  | 18.43 | 11.33 | 0.106 | 3.13 | 7.07 |
|  | 26.90 | 16.54 | 0.119 | 3.90 | 5.56 |
|  | 25.77 | 20.22 | 0.094 | 3.41 | 4.43 |
|  | 28.24 | 16.45 | 0.113 | 5.15 | 5.89 |
|  | 38.16 | 16.41 | 0.106 | 3.38 | 4.17 |
|  | 11.72 | 7.70 | 0.091 | 3.29 | 3.68 |
|  | 14.28 | 8.12 | 0.119 | 2.83 | 4.83 |
|  | 27.58 | 11.96 | 0.125 | 4.27 | 5.83 |
| post lesion | 24.03 | 16.41 | 0.113 | 5.10 | 7.13 |

**Appendix 1—key resources table**

| Reagent type (species) or resource | Designation | Source or reference | Identifiers | Additional information |
|---|---|---|---|---|
| Strain, strain background (*Danio rerio*) | Tg(aldoca:TRPV1-TagRFP) | This study | N/A |  |
| Strain, strain background (*D. rerio*) | Tg(UAS:GCaMP6s) | *Thiele et al., 2014* | ZFIN: ZDB- TGCONSTRCT-140811–3 |  |
| Strain, strain background (*D. rerio*) | Tg(aldoca:GAL4) | *Takeuchi et al., 2015* | ZFIN: ZDB- TGCONSTRCT-150414–2 |  |

*Appendix 1 Continued on next page*

*Appendix 1 Continued*

| Reagent type (species) or resource | Designation | Source or reference | Identifiers | Additional information |
|---|---|---|---|---|
| Strain, strain background (*D. rerio*) | Tg(elavl3:h2B-GCaMP6f) | *Dunn et al., 2016* | ZFIN: ZDB- TGCONSTRCT-150916–4 | |
| Chemical compound, drug | Low melting point agarose | Thermo Fisher Scientific | 16520 | |
| Chemical compound, drug | Methylene blue | Sigma Aldrich | M9140 | |
| Chemical compound, drug | Capsaicin | Sigma Aldrich | M2028 | |
| Software, algorithm | Fiji/ImageJ | *Schindelin et al., 2012* | RRID:SCR_002285 | |
| Software, algorithm | Adobe Illustrator (2020) | Adobe | RRID:SCR_010279 | |
| Software, algorithm | Matlab 2020b | Mathworks | RRID:SCR_001622 | |

