## [Editor Report · eLife Assessment]

This **important** study successfully applies an innovative chemogenetic tool to investigate cerebellar function to advance our understanding of the contributions of Purkinje cell populations to postural control in larval zebrafish. The evidence supporting the conclusions is **convincing** and supported by rigorous statistical analysis. The study highlights the power of combining genetically targeted perturbations with quantitative high-throughput behavioral analysis and original microscopy tools.

---

## [Referee Report · Reviewer #1 (Public review)]

This study uses a variety of approaches to explore the role of cerebellum, and in particular Purkinje cells (PCs), in the development of postural control in larval zebrafish. A chemogenetic approach is used to either ablate PCs or disrupt their normal activity and a powerful, high-throughput behavioural tracking system then enables quantitative assessment of swim kinematics. Using this strategy, convincing evidence is presented that PCs are required for normal postural control in the pitch axis. Calcium imaging further shows that PCs encode tilt direction. Evidence is also presented that suggests the role of the cerebellum changes over the course of early development, although this claim is less robust. Finally, the authors build on their prior work showing that both axial muscles and pectoral fins contribute to "climbs" and show convincing evidence that PCs are required for speed-dependent engagement of the fins during this behavior. Overall, establishing a role for cerebellum in postural control is not very surprising. However, a clear motivation of this study was to establish a robust experimental platform to investigate the changing role of cerebellar circuits in the development of postural control in the highly experimentally accessible zebrafish larvae and in this regard the authors have certainly succeeded.

This revised version of the manuscript incorporates several improvements. In particular, additional analysis and methodological detail is provided regarding the chemogenetic manipulation, there is expanded analysis of the speed-dependency of pectoral fin engagement, and aspects of the decoding analysis are clearer. However, it is still not certain that the emergence of a dive phenotype over development (from 7 to 14 day post fertilisation) really represents changing role for the cerebellum as opposed to changing sensitivity of Purkinje cells to the chemogenetic treatment.

---

## [Referee Report · Reviewer #2 (Public review)]

Franziska Auer et al. successfully applied the TRPV1/capsaicin tool to study the contribution of Purkinje cells to postural control. They leveraged the ability of this tool to both activate and ablate neurons within the same construct and tested its effects using their smart, high-throughput behavioral setup for postural control monitoring. With Purkinje cells ablated, balance did not appear to be disrupted; however, postural control was clearly modified along the pitch axis, with larval zebrafish maintaining, on average, a more nose-down posture compared to controls. While this effect is subtle, it is statistically robust and consistent with the group's previous findings using KillerRed-mediated ablation of Purkinje cells, where the observed postural angle change was explained by a disruption in cerebellar-mediated fin-trunk coordination. Here, the authors present a novel insight, demonstrating that this coordination is swim-speed dependent.

Furthermore, the authors convincingly activated Purkinje cells at 7 dpf, and reported modifications in posture pitch angle comparable to those observed when ablating Purkinje cells. The authors suggest a potential desynchronization of Purkinje cells to explain this observation. Future characterization and application of this activation method to other developmental time points could be of major interest. The authors successfully validated the transfer of the TRPV1/capsaicin method for targeted cell ablation and activation to the study of cerebellar functions and reinforced our current understanding of the role of Purkinje cells in postural control.

This study also explores the developmental evolution of cerebellar function in postural control by comparing the effects of Purkinje cell ablation at 7 dpf and 14 dpf. Interestingly, only dive bout posture showed differential effects across these time points, with no significant impact at 7 dpf but a significant change in postural pitch angle at 14 dpf. In contrast, the effect of Purkinje cell ablation on the climbing bout postural angle remained comparable at both ages. Including additional developmental time points would further strengthen this critical characterization of cerebellar maturation in the context of postural control.

To examine whether Purkinje cell activity encodes postural tilt angle, the authors performed calcium imaging on 31 cells from 8 fish using their Tilt In Place Microscope (TIPM). They found that tilt-angle could be decoded from individual neurons with highly tuned responses, as well as from neurons that were not obviously tuned when pooling their data. The authors refer to this effect as pseudo-population coding because recordings were performed non-simultaneously across animals.

This study successfully integrates cutting-edge genetic tools, high-throughput behavioral assays, and advanced optical microscopy to investigate the role of populations of Purkinje cells in postural control. The authors have not only validated these powerful tools but have also provided novel insights into the cerebellar involvement in postural control, including the swim-speed dependence of fin-trunk coordination.

This work represents an important step toward a detailed understanding of cerebellar contributions to postural control and highlights the potential of combining genetically targeted perturbation with quantitative behavioral analysis.

The authors have addressed my previous concerns, and I congratulate them for their excellent work.

---

## [Referee Report · Reviewer #3 (Public review)]

Summary:

This paper uses a new chemogenetic tool to investigate the role of cerebellar Purkinje cells in postural control. Using a high-throughput behavioral assay, they show that activation or ablation of Purkinje cells affects various aspects of postural control in zebrafish larvae during spontaneous swimming, and that the effects are more pronounced at later developmental time points, where the Purkinje cell number is much greater. Using a sophisticated imaging assay, they record Purkinje cell activity in response to tilt of the fish, and show that some Purkinje cells are tuned to tilt direction, and that the direction can even be decoded from untuned neurons.

Strengths:

Overall the study is nice, using a variety of genetic tools and behavioral analysis to address a fundamental question about the role of the cerebellum in postural control in fish

---

## [Author Response]

The following is the authors’ response to the original reviews.

**Public Reviews:**
**Reviewer #1 (Public Review)**:This study uses a variety of approaches to explore the role of the cerebellum, and in particular Purkinje cells (PCs), in the development of postural control in larval zebrafish. A chemogenetic approach is used to either ablate PCs or disrupt their normal activity and a powerful, high-throughput behavioural tracking system then enables quantitative assessment of swim kinematics. Using this strategy, convincing evidence is presented that PCs are required for normal postural control in the pitch axis. Calcium imaging further shows that PCs encode tilt direction. Evidence is also presented that suggests the role of the cerebellum changes over the course of early development, although this claim is rather less robust in the current version of the paper. Finally, the authors build on their prior work showing that both axial muscles and pectoral fins contribute to "climbs" and show evidence that suggests PCs are required for correct engagement of the fins during this behaviour. Overall, establishing a role for the cerebellum in postural control is not very surprising. However, a clear motivation of this study was to establish a robust experimental platform to investigate the changing role of cerebellar circuits in the development of postural control in the highly experimentally accessible zebrafish larvae, and in this regard, the authors have certainly succeeded.Overall, I consider this an excellent paper, with some room for improvement in aspects of presentation, discussion, and some aspects of the data analysis..

We thank the reviewer for their kind comments and support. In the revision we have addressed their concerns regarding data presentation and analysis. Additionally, we have expanded our introduction and discussion to address questions of presentation.

**Reviewer #2 (Public Review):**
Summary:Franziska Auer et al. investigate the role of cerebellar Purkinje cells in controlling posture in larval zebrafish using the chemogenetic tool TRPV1/capsaicin to bidirectionally manipulate (i.e., activate or ablate) these cells. This tool has been developed for zebrafish previously but has not been applied to Purkinje cells.High-throughput behavioral experiments are presented to monitor how body posture is affected by these perturbations. The analysis of postural control focuses on a specific subaspect of posture: the body tilt-angle relative to horizontal just before a swim bout is executed, quantified separately for pre-ascent and pre-dive bouts. They report a broad bimodal distribution of pre-ascent bout posture ranging from -20 to +40 degrees, while the pre-dive bout posture was more Gaussian, ranging between -40 and 0 degrees. The treatment effect is quantified as the change in the median of these distributions.Purkinje cell activation and ablation in 7 days post-fertilization (dpf) fish shifted the median of the ascending bout posture distributions to positive values. The authors hypothesize that the stochastic nature of the activation process might desynchronize Purkinje cell activity, thus abolishing Purkinje cells' role in postural control, similar to ablation. However, this does not explain why dive bout posture decreased upon activation but was unaffected by ablation.To test whether the role of Purkinje cells in postural control matures over development, the authors repeated the ablation experiments at 14 dpf. They state that "at 14 dpf, the effects of Purkinje cell lesions on posture were more widespread than at 7 dpf." However, this effect size is comparable to that observed at 7 dpf, suggesting no further maturation of the role of Purkinje cells in pre-ascending bout postural control. The median pre-dive bout posture decreased at 14 dpf, contrasting with no effect at 7 dpf, yet this change was comparable in effect size to the activation effect on Purkinje cells at 7 dpf. The current data breadth may not be sufficient to conclude that signatures of emerging cerebellar control of posture across early development were uncovered.The study's exploration of activating Purkinje cells in freely swimming fish using TRPV1/ capsaicin is of special interest, but the practicability of this method is unclear from the current presentation. It would be beneficial to present the distribution of the percentage of activatable Purkinje cells across animals and time points to provide insight into the method's efficiency. Discussing this limitation and potential improvements would aid in evaluating the method, especially since the authors report that the activation experiments were labor-intensive, limiting repeat experiments. This may explain why the activation experiment at 7 dpf is the only data presented with cell activation, with other analyses performed using the cell ablation capabilities of the TRPV1/capsaicin method.Another data point at 14dpf would significantly strengthen the conclusions.The authors analyze Purkinje cell-controlled fin-trunk coordination by examining ascending bout posture across different swim bout speeds. They make the important finding that pectoral fin movements contribute significant lift for median and fast swim bouts but not for slow ones, and that Purkinje cell ablation disrupts lift generation at all speeds.Finally, the authors examined whether Purkinje cell activity encodes postural tilt-angle by performing calcium imaging on 31 cells from 8 fish using their Tilt In Place Microscope (TIPM). They report that they could decode the tilt-angle from individual neurons with a highly tuned response, and also from neurons that were not obviously tuned when pooling them and analyzing the population response. However, due to the non-simultaneous recordings across animals, definitive conclusions about populationlevel encoding should be made cautiously, it might be better to suggest potential population encoding that needs confirmation with more targeted experiments involving simultaneous recordings.Strengths:- The study introduces a novel application of the chemogenetic tool TRPV1/capsaicin to study cerebellar function in zebrafish.- High-throughput behavioral experiments provide detailed analysis of postural control.- The further investigation of Purkinje cell-controlled fin-trunk coordination offers new insights into motor control mechanisms.- The use of calcium imaging to decode postural tilt-angle from Purkinje cell activity presents interesting preliminary results on neuronal population encoding.Weaknesses:- The term "disruption" for postural control effects may lead to misleading expectations.- The supporting data show only subtle median shifts in postural angle, raising questions about the significance of observed effects. Statistical methods that account for the hierarchical structure of the data might be required to support the conclusions.- The study's data breadth may not be sufficient to conclude emerging cerebellar postural control across early development.- The current presentation does not adequately detail the practicability and efficiency of the TRPV1/capsaicin method for activating Purkinje cells, and the labor-intensive nature of these experiments constrains the ability to replicate and validate the findings.- Non-simultaneous recordings in calcium imaging necessitate cautious interpretation of population-level encoding results.

We appreciate the reviewer's thoughtful and detailed feedback. In response, we have made several changes to highlight key points in our manuscript. We have adjusted our wording to more accurately reflect the scope of our findings. Finally, we have clarified and expanded the methods used.

**Reviewer #3 (Public Review):**
Summary:This paper uses a new chemogenetic tool to investigate the role of cerebellar Purkinje cells in postural control. Using a high-throughput behavioral assay, they show that activation or ablation of Purkinje cells affects various aspects of postural control in zebrafish larvae during spontaneous swimming and that the effects are more pronounced at later developmental time points, where the Purkinje cell number is much greater. Using a sophisticated imaging assay, they record Purkinje cell activity in response to the tilt of the fish and show that some Purkinje cells are tuned to tilt direction and that the direction can even be decoded from untuned neurons.Strengths:Overall the study is nice, using a range of tools to address a fundamental question about the role of the cerebellum in postural control in fish.Weaknesses:(1) The data in Figure 1 that establishes the method seems to be based on a very small number of experiments and lacks some statistical analysis.(2) The choice and presentation of the statistical and analysis methods used in Figures 2-5 could be improved.

We thank the reviewer for their comments. We have added additional statistical analyses for the activation experiments, and improved data presentation .

**Recommendations for the authors**:
**Reviewer #1 (Recommendations For The Authors):**
Overall I think this is a great paper.* Introduction and Discussion.The Introduction (and Discussion) do little to explain what is understood about cerebellar control of posture and what major outstanding questions remain. The first paragraph of the Introduction seems to argue that the role of the cerebellum in control of posture is well established and line 24 attempts to motivate the present study by virtue of the fact that terrestrial locomotion is "complex". This might be true but is not necessarily a major obstacle given the suite of powerful approaches available in rodent neuroscience. What are the major challenges that are hard to tackle in rodents and what specific questions can the larval zebrafish help to answer? What about development (which gets no mention at all)? I'm not suggesting a comprehensive review of every aspect of cerebellar physiology, but I think the Introduction should attempt to outline the current hypotheses in a little more detail and highlight what we still need to understand.

We take the Reviewer’s point that there is more to say in the Introduction. We feel that multi-dimensional limb biomechanics and proprioception are two aspects of terrestrial locomotion that support our use of the word “complexity.” However, we don’t dwell on this point because, as the reviewer correctly states, the suite of tools for rodent neuroscience & behavior is expansive and, in our opinion, not a limiting factor. Instead, we said what we felt we could regarding the potential contribution of the larval zebrafish in the last paragraph of the Discussion. In the revision, we have added details about the development of cerebellum to the introduction (though this, of course, is an expansive topic and well-beyond the scope of the Introduction), highlighted some of the historical limitations in rodent posture analysis, and set up the .

* Figure 2: 'Arrows denote the shift towards more nose-up postures'. I think the distribution is quite easy to interpret without these arrows; I suggest removing them.

We have removed the arrows.

* IQR is sometimes stated as a single number and sometimes as a range. It should be consistent and unless eLife has guidance to the contrary, I suggest that it be the latter.

Thank you for pointing that out. We now report it as the value at the 25&75th %ile for all IQRs.

* Figure S2: For 14 dpf fish the axes are labelled PC2/3 - is this an error?

We have changed it to a 3-dimensional plot for both 7 and 14 dpf data to show comparable plots for both ages (now Figure S5 F and G). For the analysis in the 14dpf fish the clearest separation was in the space defined by the 2nd and 3rd principal component.

* In the methods, there is insufficient detail given about fluorescent imaging.

We added additional information to how the fluorescent imaging was performed to the ‘Confocal imaging’ section as well as to the ‘Functional imaging section’

* AbstractIn my opinion, the statement "Here, we used a powerful chemogenetic tool (TRPV1/ capsaicin) to *define the role of Purkinje cells*..." is too strong. Whilst the evidence that PCs are required for postural control is certainly strong, what exactly these cells do in the service of postural control is far from clear (as the authors indeed acknowledge in the Discussion). As such, I wouldn't say their role has been "defined".

We change the word to “describe” to better reflect our findings

* aldoca transgenic.This appears to be a beautiful transgenic line but the data showing the extent of its expression and evidence that in the cerebellum it exclusively labels PCs isn't clear enough.(i) Ideally Figure 1A would show an image of a whole animal to provide an overview of transgene expression but instead it seems to be (the legend is unclear) a cartoon with a confocal projection of part of the brain overlaid.

We have updated the figure legend to be clearer that we show a cartoon of a larval zebrafish with the confocal image overlaid. The aldoca promotor has been previously described and exclusively labels Purkinje cells (10.1523/JNEUROSCI.3352-10.2010)

(ii) Figure 1B shows expression in the cerebellum, but how are we to understand that all the labelled cells are PCs? Are all PCs labelled, or only a subset? Perhaps a double labelling with a PC in situ marker could be done to demonstrate colocalisation?

As above, the aldoca promotor has been previously described; to the best of our knowledge in the Hibi lab’s hands (and ours) it labels Purkinje cells exclusively, and it labels all of them (10.1523/JNEUROSCI.3352-10.2010)

* Chemogenetic validation.Overall, the chemogenetic approach to abrogate PC function looks to be very powerful. The authors state in several places that a contribution of this paper is in its "establishing the validity of TRPV1/capsaicin-mediated perturbations". However, the data in Figure 1, along with various comments in other parts of the paper raise some questions:(i) For experiments depolarising PCs with 1µM CSn, the same size is tiny: Two transgenic animals and one control. Moreover, it is stated 'in one fish ... we observed a small number of neurons at the 9h timepoint with bright, speckled fluorescence suggestive of cell death". Was this one out of two transgenics?! In the discussion, I didn't understand the statement "ensure adequate brightness levels *to achieve sufficient depolarization without excitotoxicity*". Does this "excitotoxicity" relate to the specked fluorescence observation?Overall, the very small sample size and comments about excitotoxicity and cell death raise concerns about the approach that I think warrant clearer treatment in the results (including information about the assessment of transgene expression, % embryos judged to have suitable expression), especially as this paper is seeking to establish the validity of the method.

We note first that the method has been previously validated (https://doi.org/10.1038/nmeth.3691) and that we build on this work. For the experiment described, the point was to identify an acceptable duration for exposure. To that end, we analyzed 6 animals for up to 6h (including the washout experiments in Figure S1B) where we never observed any speckled fluorescence; we limited our behavioral experiments to 6h accordingly. We thought it would be worth including the observation of speckled fluorescence at 9h timepoint for future reference. To directly address the comment we have increased the number of analyzed cells and fish for the 1uM capsaicin experiments and added statistical analysis (lines 65-67).

When screening for transgene expression we selected for fish that had clearly visible expression, but that did not look overly bright, and used the same criteria when screening fish for the GCaMP imaging and for behavior. Around a quarter of the fish that had aldoca:TRPV1-tagRFP expression had a usable expression level for the activation experiment. We have added this information to the Results (line 62) and Methods (line 369-372)

(ii) The authors note "capsaicin could sporadically activate subsets of Purkinje cells" and further speculate about PC activity and synchrony in the discussion. Figure 1 seems to rely on single images at widely spaced time points but given that they are set up to do 2-photon calcium imaging, why didn't they collect continuous time series data and analyse the temporal patterns of activity across the transgenic PC population?

We have added time series data for calcium imaging after 1uM of Capsaicin in TRPV1- and TRPV1+ cells to Supplementary Figure S1A. Here too we see sporadic increases in calcium levels at similar rates: 0% for TRPV1- and 15-19% for TRPV1+ (see also Figure S1 legend)

(iii) The axonopathy and cell death resulting from 10 µM Csn is quite dramatic.However, here the authors do not appear to have included a TRPV1 negative control (although oddly they did for 1 µM treatment) so it is currently unclear whether or not a high conc of Csn alone might be cytotoxic.

Chen et al (https://doi.org/10.1038/nmeth.3691) have established the TRPV1/capsaicin method in zebrafish with broad neuronal label and did not see any effect with high doses of capsaicin in TRPV1 negative fish.

* Behavioural assessment - statsOverall, the disruption of postural stability after PC manipulations is convincing.However, I have a few queries about the statistics:(i) In this section, the statistical unit was not clear. The tables, which are otherwise very useful, give no indication of N. The legend text does report "8 repeats/149 control fish" and "across experimental repeats" suggesting the statistical unit might be the repeats rather than animals, but this should be clarified. In Figure 2G, individual data points should be plotted if N=8, or a representation of the distribution (eg violin or box and whisker plots) if N = 149.

We apologize for the confusion. Given the variable numbers of bouts, a single experimental repeat does not allow for an accurate estimate of expected value. Below we simulated how accurately the median can be estimated based on increasing sample sizes (Author response image 1). Given that large numbers of bouts are necessary to accurately estimate the median we pool the data for all experiments and use resampling statistics to estimate bias in our estimate.

**Author response image 1. sa4fig1:** Median estimation based on increasing sample size.

(ii) Related to the above, I hope it might be easier to interpret the unexpected change in climb posture in ablation controls once the data for individual repeats is shown.

When we analyze the data as single repeats we see considerable variability between different repeats due to undersampling. We tested the medians for the single repeats for outliers to ensure that the shift is not due to a single repeat skewing the distribution. We did not detect any outliers in the pre-lesion control or in the post-lesion control group. (Outliers were determined as deviating more than 3 times the scaled median absolute deviation (MAD) from the median. A scaling factor of 1.4826 was used to ensure that MAD-based outlier detection is consistent with other methods like Z-scores.) We added this information to line 133-134 and the method section under Statistics.

(iii) In some parts of this section, including the Tables, the authors report the 95% CI of the median, rather than IQR. In this case, they should report the z-value used for 95% CI estimation.

As we are using resampling to estimate the 95% confidence interval of the median there is no z-value as in a traditional normal distribution based confidence interval; Instead, we explicitly define the 2.5th and 97.5th percentiles from the bootstrapped sample distribution, which captures the middle 95% of the data, representing the 95% confidence interval.

* It is stated that "fish adopted more nose-up postures before *and throughout* climb bouts". Figure 2F seems to show posture before the climb, but where is the "throughout" data? It would be useful if Figure 2E, J could be extended to make a bit clearer these two phases of postural assessment.

We removed the phrase ‘throughout climb bouts’ as we are not showing the posture throughout the bout and to avoid over complicating the interpretation.

* Why were PCs not activated at 14 dpf (eg using 1 µM Csn)?

Due to shifts in priorities the first author will not be continuing this series of experiments, and so this additional experiment will have to wait for someone to pick up this line of inquiry

* The authors appear to claim that the difference in phenotype in 7 versus 14 dpf animals following high conc Csn treatment is indicative of a changing role for cerebellar PCs over this developmental period. For instance, in reference to the 14 dpf ablation phenotype, the authors write "reveals the functional emergence of Purkinje cell control of dives" and in the abstract they talk about "emerging control of posture across early development". However, can they rule out that the phenotypic differences might instead reflect differential sensitivity of the relevant PC (sub)populations to CSn at the two ages? If this caveat cannot be discounted then I suggest it is acknowledged e.g. in the discussion.

As previously established, all Purkinje cells are labeled in the aldoca line (10.1523/ JNEUROSCI.3352-10.2010). Fluorescence is brighter at 14dpf compared to 7dpf, suggesting higher levels of TRPV1. We therefore assume that at 14 dpf, the high concentration of Csn is sufficient to ablate Purkinje cells. At 14 dpf, cerebellar damage is visible under a standard dissecting microscope.The preponderance of evidence therefore speaks against a previously undiscovered subpopulation of TRPV1expressing Purkinje cells that are, by mechanisms yet unknown, resistant to high doses of capsaicin.

* Fin-body "coordination"The ideas and data around fin-body coordination are very intriguing.(i) The statement "fin engagement is speed-dependent" would benefit from a stats test to show this is indeed significant. The data in Figure 4B suggest a rather high degree of variance.

This is an important point; we appreciate the Reviewer’s attention. We have added statistics to show this is speed dependent to line 167-169 and show the corresponding plot in the supplement in Figure S4. "Here, we observed that fin engagement is speeddependent, with faster bouts producing greater lift for a given axial rotation (Spearman correlation coefficient: control 0.2193; 10uM capsaicin: 0.0397; Z-test after ztransformation: p < 0.001)

(ii) The statement "After capsaicin exposure, the slopes of the medium fast speed bins were significantly lower (Figure 4C), reflecting *a loss of speed-dependent modulation*" is not convincing. The slope is likely a function of both speed and Csn treatment, and the comparisons in Figure 4C appear to be testing the latter, not the former.

We understand the reviewer’s point. However, the slope for the slow bouts remains unchanged. We therefore conclude that the reduction in fin-body slope is speed dependent and not a speed independent reduction of slope overall.

We have made this more clear by adding Supplementary Figure S4 and changing the text in line 177-179.

(iii) I'd like to understand more about the phenotype of the fin-amputated animals. Were any "bout" parameters changed? Did the animals still attempt climbs and was the distribution of the upward rotation parameter similar to controls? The text states "the slope of the relationship between upward rotation and lift was indistinguishable from zero" but the stats reported in the text are comparisons between groups while Table 5 shows 95% CIs that don't span zero. Some clarification would be useful here.

We appreciate the Reviewer’s interest. We’ve studied climbing in fin-amputated animals at length here: https://doi.org/10.7554/eLife.45839 and here: https://doi.org/10.1016/ j.celrep.2023.112573 and have added these references in line 183.

(iv) The authors repeatedly refer to fin-body *coordination* but it is not clear whether the loss of lift after PC ablation is a result of an explicit coordination defect (i.e. changes in the relative timing and/or kinematics between fins and axial motion components), versus a simple reduction in pectoral fin engagement. Either result could be interesting, but this should be clarified.

Thank you for pointing that out. In the fastest speed bin, we observed an increase in upward rotation and a decrease in average fin lift. In contrast, the medium speed bin showed no significant changes in average fin lift or upward rotation (see Author response image 2 and Tables 4 and 5), yet already displayed coordination deficits. Based on these observations, we argue that Purkinje cell lesions primarily affect coordination, rather than simply reducing one specific parameter such as lift or rotation (line 293-298).

We have added fin lift and rotation values from Author response image 2 for all speed bins to tables 4 and 5.

**Author response image 2. sa4fig2:** Fin lift and rotation for slow, medium and fast bouts.

* PC activity and decoding of pitch direction.The clever TIPM method is used to collect calcium data that convincingly shows that individual PCs can encode pitch-tilt direction. However, a population of "not tuned" cells are also identified, and here I found the analysis of their responses and the argument that they encode pitch direction at a population level difficult to follow.(i) First, although the naming of the cells implies that individual neurons do not encode pitch direction, I did not find this convincing. Figures 5F/G suggest that several "not tuned" cells in fact show quite consistent differences in activity across trial types and indeed in terms of their average responses sit as far from the unity line as do several "tuned" cells.

The Reviewer’s comment helped us clarify some key points. First, tuned and untuned cells were categorized based on a Directionality Index threshold of 0.35; some cells might look similar in 5F/G but the highly variable responses of Purkinje cells have highly variable response so overall there was no consistent tuning. We have clarified this in the text in line 203-207 Below we have plotted the Up versus Down responses for the 10 least tuned cells (sorted by directionality index). While some cells have higher responses on average to one direction we think that the variability makes it difficult to support a claim for “tuning.” We have also tested the support vector machine on the least tuned cells to confirm that the chosen cutoff for tuned/untuned is not affecting our claim that untuned cells can encode position.(see also Author response image 4)

**Author response image 3. sa4fig3:** Trial-by-trial variability.

(ii) It is therefore not very surprising that PCA (and the SVM decoder) distinguishes trial type. I would guess that PCA assigns the largest weights to these most tuned of the "not tuned" cells, and the 3-5 cell decoders do well when these cells happen to be sampled.

**Author response image 4. sa4fig4:** Decoding accuracy of the 3/5/7 least tuned cells.

This was an interesting idea. To rule out that it is only the most tuned cells that contain the information, we tested the decoder on the 3/5/7 least tuned cells; here too, 5 and more cells are better able to accurately decode the direction. We have add the decoding accuracy to the text in line 221-224

(iii) As I understand the analysis, Figure 5G shows responses for "not tuned" cells over 21 trials (of each type) but these are not the same trials for the different cells? How then is population coding being assessed?

We have updated the text and refer to this data as a “pseudo-population” in lines 216 and 218 for all experiments where we combined cells from different fish. For technical reasons, when we perform TIPM at eccentric angles we must use sparsely labelled fish to ensure that we can find the same cells over a 60 degree range. We have repeated our analyses for TIPM centered at the horizon, where we can record from entire populations from a single fish.

(iv) Furthermore, Figure S2 shows a somewhat different analysis with decoding accuracy measured on a fish-by-fish basis. In this case, are these decoders for simultaneously imaged neurons? Is this a cross-validated measure of decoding accuracy?

Yes, as above, Figure S4 (former S2) looks at fish-by-fish basis of simultaneous recorded neurons. Yes, it was 5-fold cross validated. We have updated the text in line 490-494.

**Reviewer #2 (Recommendations For The Authors):**
- Postural control involves various aspects such as balance, coordination, relative body part orientations, and stability. Discussing these and presenting in this context the specific subaspect characterized in this study would help clarify which aspect of postural control the work focuses on.

The Reviewer makes an interesting point, but we think their description of what constitutes postural control is overly broad. Specifically, control of “relative body part orientations in space” by definition requires coordination, and subserves balance and stability. We acknowledge, of course, that different aspects can be and often are treated independently. While interesting, a full treatment of what comprises “postural control” is beyond the scope of the paper, as it would require reconciling the terms across taxa, effectors, environments and well over a century of experiments.

We contend that posture — particularly underwater — is best defined as the relative orientation of body parts in space. For fish, those parts consist of predominantly axial muscles and secondarily fins. We present these definitions in the Introduction and thank the Reviewer for encouraging us to more clearly shape our findings.

- Disruption of posture or postural control: The use of the word "disruption" could lead to misleading expectations. While it may not be incorrect, it suggests a significant loss of equilibrium, an obvious increase in postural variability, or at least a noticeable effect when observing an individual animal's behavior. However, the supporting data show only a subtle median shift in postural angle within a very broad distribution averaged over many individuals. This effect was only significant when comparing fish with a control group, not when comparing fish posture before and after the treatment.Replacing "disruption" with "modification" would be more cautious.

We take the Reviewer’s point and have adjusted our wording to "modifies postural control.” In lines 137, 266, and 283

- Statistical significance: Consider aligning the asterisk notation with conventional standards (e.g., * for p < 0.05, ** for p < 0.01, *** for p < 0.001) to enhance clarity for readers. On the other hand, the individual measurements might not be independent (e.g., measurements from the same fish, or the same tank are likely to be correlated), so using the Wilcoxon rank-sum test (Mann-Whitney U test) on pooled data might lead to incorrect conclusions. Methods that account for the hierarchical structure of the data might be required to support the conclusions.

We take the Reviewer’s point about the importance of conventions, however we have never found “more stars = more significant” to be all that helpful in evaluating claims. Instead, we’ve opted to have both a significance and effect size criteria; a “star” here reflects our considered confidence in the difference we observe.

We agree that the hierarchical nature of pooled data is worth considering/presenting.

We performed a two-way analysis of variance (ANOVA) on the interquartile ranges (IQRs) of the single experimental repeats for the 7 days post-fertilization (dpf) activation, 7dpf lesion, and 14dpf lesion experiments. The ANOVA revealed no significant main effects, supporting the strategy of pooling experimental repeats to estimate distributions.

The results of the ANOVA, along with the IQRs for all experimental repeats, are presented in Tables 6-11. We have also clarified this in the methods section in lines 505-509.

- Data representation: All data of postural angles should be represented in the form of violin plots to show the underlying distributions of the postural angles, especially given that the effect size is small relative to the dispersion of the distribution of the postural angle and that this distribution is also not Gaussian but bimodal, and different before and after the treatments.

We take the Reviewer’s point that seeing the full distribution can be useful. We have added plots of the raw distributions for the data in Figure 3 as supplemental Figure S3.

- Showing the distributions will provide the necessary information for the reader to evaluate the importance of the effect. For all data shown in Table 1, the distributions should be presented in the supplementary information.

As requested, we have added the distributions of the data in Table 1 to the supplement (Figure S2)

- Roll posture: A statement about whether roll posture is perturbed by Purkinje cell manipulation would be a piece of important additional information helping to understand how strong the 'disruption' of posture is.

We haven’t assessed roll posture, as this is not practical in the current version of the SAMPL apparatus. We have added this limitation to the results (line 116) but also note that as our manipulations are bilateral, we don’t anticipate any systematic changes to roll.

- Comparison with other methods: Add a discussion on how the TRPV1/capsaicin method compares with other methods, such as using nitroreductase (Ntr) for targeted pharmaco-genetic ablation of cells by treatment with metronidazole or the the possibility to to ablate Purkinje cells by KillerRed as the author lab has done previously. Both methods have been applied to ablate Purkinje cells in larval zebrafish. What are the advantages of the TRPV1 method compared to these when neglecting the activation possibility?

Thank you for that suggestion, we have added a section to the discussion where we compare the TRPV1/capsaicin lesion to other lesion methods (lines 334-336)

- Describe the decoding algorithm: The decoding algorithm used could be described more in detail in the methods section.

We have described the decoding algorithm in more detail in the methods under ‘Functional GCaMP imaging in Purkinje cells.’ Line 488+

We used a support vector machine (SVM) with a linear kernel. The SVM model was trained using k-fold cross-validation, which splits the data into k subsets (folds). At each iteration, the model was trained on k-1 folds and tested on the remaining fold, ensuring that the model performance was evaluated on unseen data in each fold. Permutations were performed on randomized trial identity as a null hypothesis (5-fold cross-validation; 100 shuffles for randomization). Accuracy was calculated as 1 minus the classification loss.

- Availability of code: The link to the data and code repository is not working.

Thank you for pointing that out, we have fixed it now. In the lower right of the page you can see the history of all changes to the repository, including the entry on 2023-09-08 where the corresponding author set it to “public.” When we checked thanks to your comment, it had been set to “private,” without any record of when/why. We have reset it 2024-10-17. We will continue to check it periodically in the future and apologize in advance if it is unavailable; this is the first time we’ve seen that happen.

- Electrophysiological Control: Including an electrophysiological characterization of the activation of Purkinje cells by the TRPV1/capsaicin would significantly strengthen the validity of the method.

We take the Reviewer’s point that electrophysiological characterization is a way to strengthen the validity of the method. However, Chen et al (https://doi.org/10.1038/nmeth.3691) have performed electrophysiology during neuronal activation and concluded that TRPV1 activation with capsaicin indeed increases neuronal activity and firing rates increased. Our calcium imaging and lesion experiments amply demonstrate that Purkinje cells are sensitive to TRPV1-mediated currents. We therefore do not believe that the additional information gained by arduous electrophysiological evaluation is merited here.

- Describe more in detail how climb and dive bouts are defined. The height difference between consecutive bouts measured 250ms before the bout of executions.

Climb and Dive bouts are split by the angle of their trajectory. If the fish moves up (i.e. trajectory larger 0) it is considered a climb bout and vice versa for dive bouts. 250ms prior to the maximum speed is roughly the time the fish initiate a bout, so the pre-bout posture is measured when at this point. The time-courses of bouts are dissected extensively in Zhu et. al. 2023. We have added a definition for climb and dive bouts to the method section under ‘Behavior analysis’ line 453 and 454.

- Figure 1H: Why can't you ablate all Purkinje cells but only about 80%?

This is an excellent question. We opted for an extremely conservative count, and included everything that was still resembling a cell, even if it might not be functional/ already dying. Our counts are therefore likely an underestimate of the percentage of cells that were lost. We have added this point to the text in lines 393 395

Figure 2C: The method is not fully clear. At 8dpf 0.1uM capsaicin is added to the chamber. At what time after the application of capsaicin did the behavioral recording start?

We recorded after about 10-15min after adding the 1uM Csn to the chambers. The fish were fed after the 6h in capsaicin. We have added this information to the method section line 404 - 408.

- Figure 2F: What indicates the shown confidence interval? Also median with a 95% confidence interval calculated over the experiments in parallel?

The distributions shown in Figure 2F take data from all experiments pooled. We use resampling methods to determine the variability in our estimates. The distribution plots are showing the median and the 25th and 75th percentile of the resampled distribution. We have added this information to the figure legends.

- Figure 3: Subtitles on panel D and E indicating and would facilitate reading.

We have added the subtitles to those panels.

- Figure 4: Describe in the methods how recordings from individual fish were mapped onto each other to superimpose the Purkinje cell locations recorded from the 8 fish.

We have added the respective section to the methods: Line 481 - 483

“To map the anatomical locations of the recorded cells, we imaged overview stacks for each fish. These stacks were manually aligned in Illustrator, and the cells included in the analysis were reidentified and color-coded according to their tuning properties.”

**Reviewer #3 (Recommendations For The Authors):**
Major points:(1) Lines 74-81. The data presented here and in later experiments to argue for an effect of capsaicin on neural activity lacks statistical rigor because of the apparently very small numbers of animals/cells assessed. For example, the control appears to involve 4 cells assessed from 1 animal, and the experimental group is just 2 animals. Given that the interpretation of the paper depends upon this result, it is worthwhile to show the result more clearly, and with some statistical analysis. They argue in the discussion that "Our imaging assay established that 1 µM of capsaicin would stochastically activate subsets of Purkinje cells" which seems a stretch from the data as presented.

We appreciate this point, which was shared by Reviewer 1. We have added more data and performed statistical analysis (line 63 - 67 as well as Figure S1A)

(2) I found the practice of sorting effects by a mixture of effect size and p-value to be a little arbitrary, although in this case, it seems likely that it identified the most relevant effects. I would have preferred to see some attempt to correct for multiple comparisons (e.g. by resampling with the identities of fish shuffled to estimate the distribution of each measurement for this population size), followed by filtering for effect size after establishing a corrected threshold for significance.

We take the Reviewer’s point, though we note that critical values for effect size and pvalue are inevitably “a little arbitrary.” We can’t do the exact analysis the Reviewer suggests as we do not measure data from individual fish for these experiments. However, we did calculate new critical p-values (added to the Tables) that account for multiple comparisons using Šidák’s method.

(3) Figure 4. The data here is a little strange in that the slope in the control condition for medium speed is given as much larger than for slow, but the data in the two cases appears largely overlapping for most of the range of behavior, only diverging for the most extreme rotations. It seems perhaps that the measurement of slope is strongly dependent on these most extreme values. The authors might want to consider the use of robust regression methods which might mitigate these effects.

This is an interesting observation and we appreciate the Reviewer’s thoughtful suggestion. We now use a robust regression method (bisquare weighting of residuals).

We have adjusted all values in lines 175 - 177 and added the regression method to the Methods section line 520.

(4) Figure 5. The 'principal component analysis' description is extremely unclear. The text says that PCA 'showed near-complete segregation of trial types' but it is not explained how this was achieved with PCA or how this was quantified. Figure panels show the data plotted using different pairs of PCs showing visual evidence of segregation. In the methods, it is stated that "We performed principal component analysis" and that "cells were used for principal component analysis and subsequent support vector machine decoding analysis". What is meant exactly by 'performed PCA'? Was PCA used in a dimensionality reduction step? And if so, how many and which PCs were chosen and why? For visualization of the separation, the authors show arbitrary pairs of PCs. Could it be better to use a method more suited to that purpose such as linear discriminant analysis?

PCA was used to define a subspace to qualitatively evaluate if different trials could be separated. Once it became clear that it could, we next trained a binary decoder on the complete dataset (i.e. no dimensionality reduction). We did not perform linear discriminant analysis as the unsupervised PCA already showed separation of trial types. We have made this clearer in lines 212 - 214.

(5) Why does the decoding analysis use only untuned cells? Isn't it equally, or more, interesting to know how well tilt can be encoded using all cells? It is unclear to me what we learn by selecting only untuned cells for this analysis (although I agree it is interesting that this does work).

We focused exclusively on untuned cells because including even a single highly tuned cell for the population coding will lead to excellent results. By using untuned cells we test if there is some directionality information that is not visible just by looking at the up/ down responses of single cells. We have made this clear in lines 217 - 218

Minor points and corrections:(1) Maybe consider losing the words 'powerful' (I think it is overused and not well defined) and 'reagent'. Reagent is normally used for something that participates in a reaction. It is a bit odd to use it to refer to a transgenic animal. Later it is called a 'tool' which seems better.

We have changed the wording and refer to it as tool for the whole paper.

(2) Figure 1D. Please use a color bar to indicate the scale.

We have added a color scale to the panel

(3) Saying that 'posture' increases is confusing, although the meaning can be inferred from the overall context and the definitions in the Methods - could Posture be capitalized to indicate a specific definition is being used rather than the general meaning?

This suggestion agrees with those made by Reviewer 2. We have changed the wording to “postural angle.”

(4) The arrowheads in Figure 2FHK are unnecessary and confusing (why are some horizontal and some vertical?).

Thank you for that suggestion, we have removed the arrowheads.

(5) Figure 3 The legend should indicate that the image is shown with an inverted lookup table.

We have updated the legend

(6) Figure 3 D and E Titles would be helpful, so it is not necessary to refer to the legend to understand the difference.

We have added titles to the figure panels

(7) The dwell time for the 2-photon experiments is given in the manuscript, but I think the authors meant microseconds?

Thank you for pointing that out. We have corrected it to microseconds.